# Perceptual error based on Bayesian cue combination drives implicit motor adaptation

**Zhaoran Zhang[1†], Huijun Wang[1†], Tianyang Zhang[1], Zixuan Nie[1], Kunlin Wei[1,2,3,4]\***

[1]School of Psychological and Cognitive Sciences, Peking University, Beijing, China; [2]Beijing Key Laboratory of Behavior and Mental Health, Beijing, China; [3]Peking-Tsinghua Center for Life Sciences, Peking University, Beijing, China; [4]National Key Laboratory of General Artificial Intelligence, Beijing, China

**Abstract** The sensorimotor system can recalibrate itself without our conscious awareness, a type of procedural learning whose computational mechanism remains undefined. Recent findings on implicit motor adaptation, such as over-learning from small perturbations and fast saturation for increasing perturbation size, challenge existing theories based on sensory errors. We argue that perceptual error, arising from the optimal combination of movement-related cues, is the primary driver of implicit adaptation. Central to our theory is the increasing sensory uncertainty of visual cues with increasing perturbations, which was validated through perceptual psychophysics (Experiment 1). Our theory predicts the learning dynamics of implicit adaptation across a spectrum of perturbation sizes on a trial-by-trial basis (Experiment 2). It explains proprioception changes and their relation to visual perturbation (Experiment 3). By modulating visual uncertainty in perturbation, we induced unique adaptation responses in line with our model predictions (Experiment 4). Overall, our perceptual error framework outperforms existing models based on sensory errors, suggesting that perceptual error in locating one's effector, supported by Bayesian cue integration, underpins the sensorimotor system's implicit adaptation.

**\*For correspondence:**
wei.kunlin@pku.edu.cn

[†]These authors contributed equally to this work

## eLife assessment

This study presents an **important** finding on the influence of visual uncertainty and Bayesian cue combination on implicit motor adaptation in young healthy participants, hereby linking perception and action during implicit adaptation. The evidence supporting the claims of the authors is **convincing**. The normative approach of the proposed PEA model, which combines ideas from separate lines of research, including vision research and motor learning, opens avenues for future developments. This work will be of interest to researchers in sensory cue integration and motor learning.

## Introduction

To achieve and sustain effective motor performance, humans consistently recalibrate their sensorimotor systems to adapt to both internal and external environmental disturbances (*Berniker and Kording, 2008*; *Shadmehr et al., 2010*; *Wolpert et al., 2011*). For instance, transitioning to a high-sensitivity gaming mouse, which drives cursor movement at an accelerated rate compared to a standard computer mouse, may initially result in decreased performance in computer-related tasks. However, humans are capable of rapidly adapting to this new visuomotor mapping within a short period of time. While conscious corrections can facilitate this adaptation process, our sensorimotor

system often times adapts itself implicitly without our conscious efforts (*Albert et al., 2021*; *Krakauer et al., 2019*).

While recent research has intensively examined the interplay between explicit and implicit learning systems (*Albert et al., 2022*; *Miyamoto et al., 2020*), several characteristics of implicit motor adaptation have emerged that challenge traditional theories. Conventionally, motor adaptation is conceptualized as error-based learning, in which learning accrues in proportion to the motor error experienced (*Cheng and Sabes, 2006*; *Donchin et al., 2003*; *Thoroughman and Shadmehr, 2000*). However, implicit adaptation exhibits an overcompensation phenomenon where the extent of adaptation surpasses the error induced by visual perturbations (*Kim et al., 2018*; *Morehead et al., 2017*). Additionally, implicit adaptation manifests a saturation effect; it increases with perturbations but plateaus across a broad range of larger perturbations (*Bond and Taylor, 2015*; *Kim et al., 2018*; *Morehead et al., 2017*; *Neville and Cressman, 2018*). These observations of overcompensation and saturation are incongruent with prevailing state-space updating models, which presuppose that incremental learning constitutes only a fraction of the motor error (*McDougle et al., 2015*; *Smith et al., 2006*). Another aspect of implicit adaptation that remains mechanistically unexplained pertains to its impact on proprioception. In traditional motor adaptation, proprioception is biased towards visual perturbation, maintaining a stable bias throughout the adaptation process (*Ruttle et al., 2016*; *Ruttle et al., 2021*). In contrast, implicit adaptation initially biases proprioceptive localization of the hand towards the visual perturbation, but this bias gradually drifts in the opposite direction over time (*Tsay et al., 2020*).

Causal inference of motor errors has been suggested to explain the discounting of large perturbations (*Wei and Körding, 2009*). However, the causal inference account predicts a decline in adaptation with increasing perturbation, diverging from the observed ramp-like saturation effect. Tsay and colleagues recently synthesized existing evidence to propose that implicit adaptation reaches an upper bound set by cerebellar error correction mechanisms, reflected in a ramp-like influence of vision on proprioception (*Tsay et al., 2022c*). While this ramp function (instantiated as proprioceptive re-alignment mode, PReMo) could explain the observed saturation, the postulate of an upper bound on visual influence lacks empirical validation. Some research supports the idea of saturation in proprioceptive recalibration (*Modchalingam et al., 2019*), yet other studies suggest a linear increase with visual perturbations (*Rossi et al., 2021*; *Salomonczyk et al., 2011*). Additionally, current models fall short of quantitatively capturing the time-dependent shifts in proprioceptive bias associated with implicit adaptation.

In this study, we put forth a unified model to account for the distinct features of implicit adaptation based on the Bayesian combination of movement-related cues. Prior models have overlooked the fact that visual uncertainty related to the perturbation increases with the size of the perturbation as the cursor moves further from the point of fixation and into the visual periphery (*Klein and Levi, 1987*; *Levi et al., 1987*). This is particularly pertinent for implicit adaptation that is widely investigated by the so-called error-clamp paradigm, in which participants are instructed to fixate on the target and disregard the perturbing cursor. Moreover, conventional theories of motor adaptation define motor error according to the sensory modality of the perturbation, that is, visual errors for visual perturbations (*Tsay et al., 2022c*; *Wei and Körding, 2009*). We propose an alternative: perceptual error drives implicit adaptation, as the sensory perturbation influences the perception of the effector's position and, subsequently, motor adaptation. Through a series of experiments, we aim to demonstrate that combining eccentricity-induced visual uncertainty (Experiment 1) with a traditional motor adaptation model (state-space model) and a classical perception model (Bayesian cue combination) can explain both over-compensation and saturation effects (Experiment 2), as well as the time-dependent changes in hand localization (*Tsay et al., 2020*). Finally, to provide causal evidence supporting our Perceptual Error Adaptation (PEA) model, we manipulated visual uncertainty and observed that subsequent adaptation was attenuated for large perturbations but not for small ones—a finding that contradicts existing models but aligns well with the PEA model. Across the board, our model outperforms those based on ramp error correction (*Tsay et al., 2022c*) and causal inference of errors (*Wei and Körding, 2009*), offering a more parsimonious explanation for the salient features of implicit adaptation.

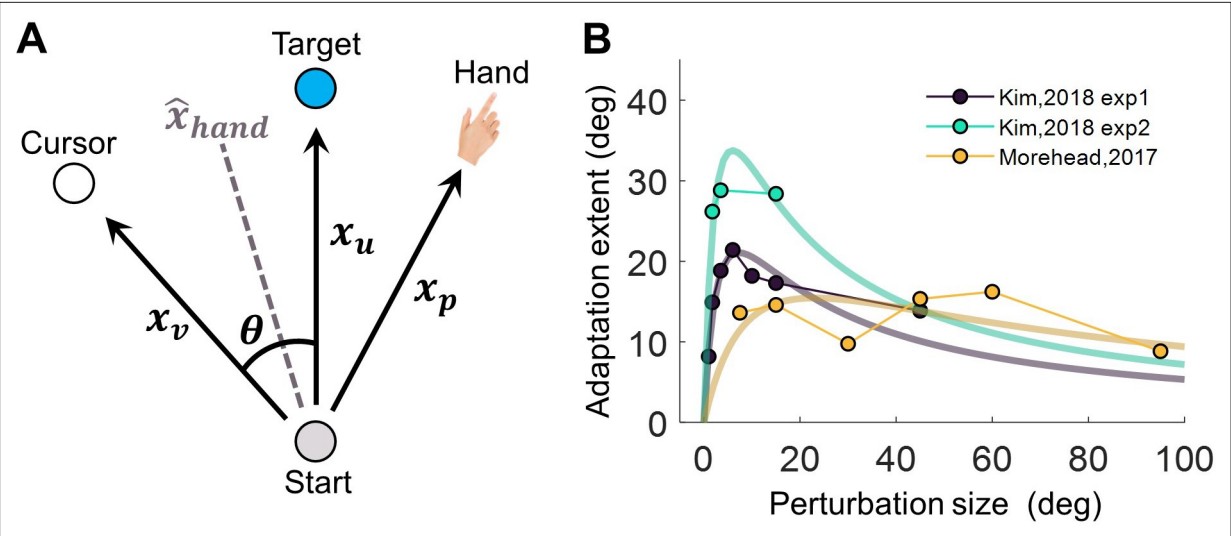

**Figure 1.** The Perceptual Error Adaptation (PEA) model for error-clamp adaptation. (**A**) Illustration of involved sensorimotor cues for estimating hand direction $\hat{x}_{Hand}$. The clamped cursor, the hand, and the sensory prediction of the reaching action provide the visual ($x_v$), proprioceptive ($x_p$), and the sensory prediction cue ($x_u$) of movement direction, respectively. The hand direction estimate is assumed to be based on maximum likelihood cue combination. (**B**) Assuming a linear dependency of visual uncertainty on eccentricity, the PEA model predicts that implicit adaptation extent is a concave function of perturbation size $\theta$, a pattern qualitatively aligning with previous findings (*Kim et al., 2018*; *Morehead et al., 2017*).

## Results

### The perceptual error adaptation model with varying visual uncertainty

We start by acknowledging that the perceptual estimation of effector position is dynamically updated and influenced by sensory perturbations during motor adaptation. For implicit adaptation studied via the error-clamp paradigm, participants are required to bring their hand to the target while ignoring the direction-clamped cursor (*Morehead et al., 2017*). Accordingly, the perceptual estimation of the hand movement direction relies on three noisy sensory cues: the visual cue from the cursor, the proprioceptive cue from the hand, and the sensory prediction of the reaching action (*Figure 1A*). Without loss of generality, we posit that each cue is governed by an independent Gaussian distribution: the visual cue $x_v$ follows $N\left(\theta, \sigma_v^2\right)$, where θ is the cursor direction and $\sigma_v^2$ is visual variance, the proprioceptive cue $x_p$ follows $N\left(x_{hand}, \sigma_p^2\right)$, where $x_{hand}$ is the hand movement direction and $\sigma_p^2$ is proprioceptive variance, and the sensory prediction cue $x_u$ follows $N\left(T, \sigma_u^2\right)$, where $T$ is the target direction and $\sigma_u^2$ is prediction variance. Participants aim for the target, expecting their hand to reach it. Using the Bayesian cue combination framework (*Berniker and Kording, 2011*), the perceived hand location ($\hat{x}_{Hand}$) on trial *n* can be derived:

$$\hat{x}_{Hand,n} = \sum_i W_i x_{i,n}, \quad with \quad W_i = \frac{1/\sigma_i^2}{\sum_j 1/\sigma_j^2}, \quad i,j = u,p,v \tag{1}$$

This estimated hand position is derived using maximum likelihood estimation from the three noisy cues. Given that the clamped cursor deviates the target by θ, the visual cue $x_v$ biases the hand estimate $\hat{x}_{Hand}$ towards the cursor's direction. This deviation from the target direction $T$ constitutes the perceptual error, which drives adaptation on the subsequent trial n+*1* (*Equation 2*). Consisting with existing models (*Albert et al., 2022*; *Cheng and Sabes, 2006*; *Herzfeld et al., 2014*; *McDougle et al., 2015*), trial-to-trial adaptation is modeled using a state-space equation:

$$x_{p,n+1} = Ax_{p,n} + B\left(T - \hat{x}_{Hand,n}\right) \tag{2}$$

where $A$ is the retention rate capturing inter-movement forgetting and $B$ is the learning rate capturing the proportion of error corrected within a trial. The interplay between forgetting and learning dictates the overall learning extent, that is, the asymptote of $x_p$:

$$x_p^{asym} = -\frac{B/\sigma_v^2}{B/\sigma_p^2 + (1-A)\sum_j 1/\sigma_j^2}\theta, \ j = v,p,u \tag{3}$$

Thus, the positive influence of perturbation size θ on the adaptation extent is counterbalanced by the rise in visual uncertainty $\sigma_v$, since sensory uncertainty of various visual stimuli increases linearly with eccentricity (*Klein and Levi, 1987*; *Levi et al., 1987*). As participants are instructed to fixate on the target, an increase in θ leads to increased eccentricity. Hence, we model this linear increase in visual uncertainty by

$$\sigma_v = a + b\theta, \tag{4}$$

where *a* and *b* are free parameters. We conducted simulations of implicit adaptation with varying error clamp size (*θ*). The model simulation closely resembles the saturated adaptation in three independent experiments (*Kim et al., 2018*; *Morehead et al., 2017*). In fact, our PEA model predicts a concave adaptation pattern, contrasting with the ramp pattern suggested by the PReMo model (*Tsay et al., 2022c*). In Experiment 1, we aim to validate the assumption of a linear increase in visual uncertainty (*Equation 1*); in Experiment 2, we seek to verify whether implicit adaptation adheres to a concave pattern as prescribed by the PEA model. Subsequent experiments, namely Experiments 3 and 4, will test the model's additional novel predictions concerning changes in proprioception and the impact of experimentally manipulated visual uncertainty on adaptation, respectively.

## Experiment 1: Visual uncertainty increases with perturbation size

To quantify visual uncertainty in a standard error-clamp adaptation setting, we employed psychometric methods. Occluded from seeing their actual hand, participants (n=18) made repetitive reaches to a target presented 10 cm straight head while an error-clamped cursor moving concurrently with one of three perturbation sizes (i.e. 4°, 16°, and 64°), randomized trial-by-trial. In alignment with the error-clamp paradigm, participants were instructed to fixate on the target and ignore the rotated cursor feedback. Eye-tracking confirmed compliance with these instructions (*Figure 2—figure supplement 1*). Perturbation directions were counterbalanced across trials, with equal probability of clockwise (CW) and counterclockwise (CCW) rotation. Post-movement, participants were required to judge the cursor's rotation direction (CW or CCW) relative to a briefly displayed reference point (*Figure 2A* and Figure 6A). Employing this two-alternative forced-choice (2AFC) task and the Parameter Estimation by Sequential Testing (PEST) procedure (*Lieberman and Pentland, 1982*), we derived psychometric functions for visual discrimination of cursor movement direction (Figure 6 and *Figure 2—figure supplement 2*). Our findings reveal a significant increase in visual uncertainty ($\sigma_v$) with perturbation size for both CW and CCW rotations (Friedman test, CW direction: $\chi^2(2)=34.11$, p=4e-8; CCW: $\chi^2(2)=26.47$, p=2e-6). Given the symmetry for the two directions, we collapsed data from both directions, and confirmed the linear relationship between $\sigma_v$ and θ by a generalized linear model: $\sigma_v = a + b\theta$, with *a* = 1.853 and *b* = 0.309, $R^2$=0.255 (*F*=51.6, p=2.53e-9). The 95% confidence intervals (CI) for *a* and *b* are [0.440, 3.266] and [0.182, 0.435], respectively. The intercept was similar to the visual uncertainty estimated in a previous study (*Tsay et al., 2021a*). We thus observed a striking sevenfold increase in visual uncertainty from a 4° perturbation to a 64° perturbation (22.641±6.024° vs. 3.172±0.453°). In addition, we observed significant correlations of $\sigma_v$ between different perturbation sizes (Spearman correlation, 4° and 16°: ñ=0.795, p<0.001; 16° and 64°: ñ=0.527, p=0.026). This finding further confirmed that the relative magnitude of visual uncertainty among individuals was consistent across perturbation sizes.

## Experiment 2: Visual uncertainty modulated perceptual error accounts for overcompensation and saturation in implicit adaptation

The critical test of the PEA model lies in its ability to employ the increase in visual uncertainty obtained from Experiment 1 to precisely explain key features of implicit adaptation. Earlier research mostly scrutinized smaller perturbation angles when reporting saturation effects (*Bond and Taylor, 2015*; *Kim et al., 2018*). In contrast, Experiment 2 involved seven participant groups (n=84) in characterizing implicit adaptation across an extensive range of perturbation sizes (i.e. 2°, 4°, 8°, 16°, 32°, 64°, and 95°). After 30 baseline training cycles without perturbations, each group underwent 80 cycles of error-clamped reaching and 10 washout cycles without visual feedback (*Figure 3A*). We replicated key

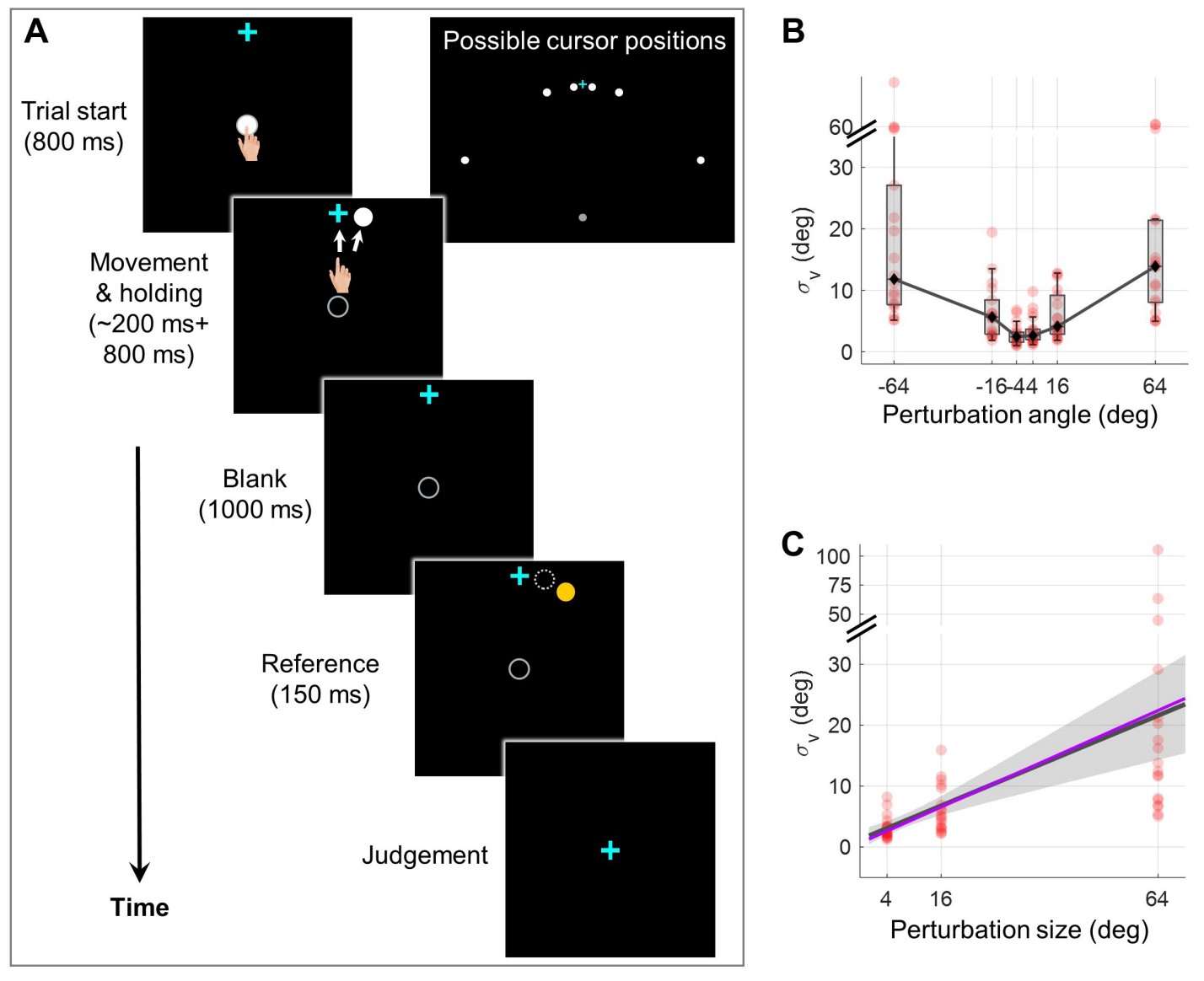

**Figure 2.** Experiment 1 measuring the dependency of visual uncertainty on perturbation size. (**A**) Illustration of all possible cursor endpoints during the experiment and the procedure of the 2AFC task for judging the cursor motion direction. In an exemplary trial, the participant reaches to a target while a direction-clamped cursor moves concurrently, serving as an error-clamp perturbation. Following a 1000 ms blank masking period, a reference point appears for 150ms, either clockwise or counterclockwise from the clamped cursor. The participant is then asked to make a binary judgment regarding the direction of the clamped cursor relative to the reference point. (**B**) The visual uncertainty, obtained from psychometrical estimation based on the 2AFC, is plotted as a function of perturbation size (n = 18). Both individual estimates (red dots) and group-level statistics (boxplots) are shown. Positive angles correspond to CW rotations, negative angles to CCW rotations. (**C**) Collapsing data from both rotation directions, we observe that visual uncertainty closely follows a linear function of perturbation size. The dark gray line and its shaded region denote the regression line and its 95% confidence interval, respectively. The purple line is generated with the values of *a* and *b* fitted from data in Experiment 2 with *a* and *b* treated as free parameters (see Methods for details).

The online version of this article includes the following figure supplement(s) for figure 2:

**Figure supplement 1.** Heat map of eye fixations during the 2AFC task in Experiment 1.

**Figure supplement 2.** Performance of an exemplary participant in Experiment 1.

features of implicit adaptation: it incrementally reached a plateau and then declined during washout. Small perturbations led to overcompensation beyond visual errors: for 2°, 4°, 8°, and 16° clamp sizes, the adaptation was substantially larger than the perturbation itself. Across perturbation sizes, the faster the early adaptation, the larger the final adaptation level (***Figure 3—figure supplement 2***). We

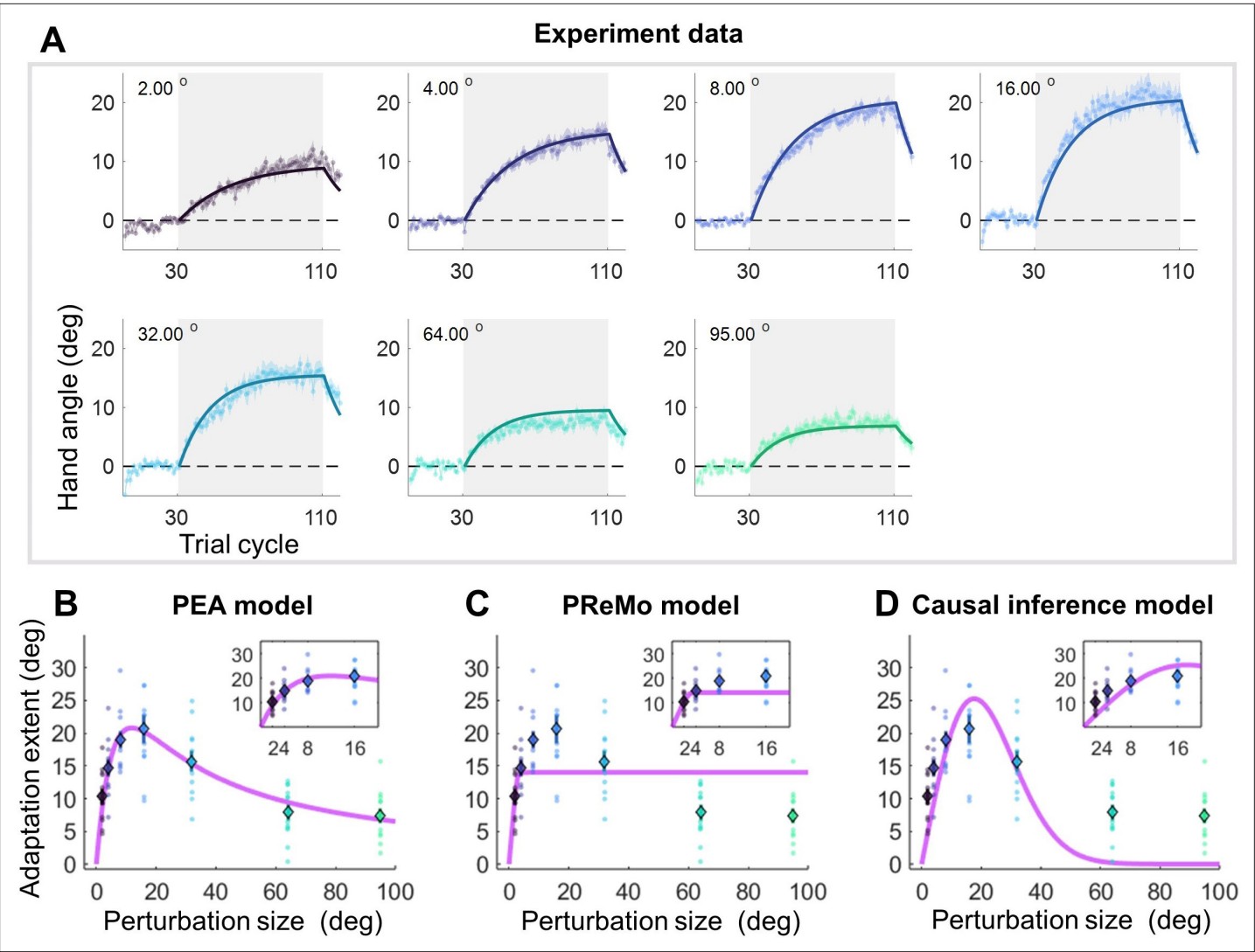

**Figure 3.** Results and model fitting of Experiment 2. (**A**) Implicit adaptation to error clamps of varying sizes is depicted (n = 84); colored dot-lines and colored shading area represent the mean and standard error for each participant group. The light gray area indicates trials with error-clamp perturbations. Adaptation starts after baseline, gradually asymptotes to its final extent, and then decays with null feedback during washout. Different perturbation sizes result in distinct adaptation rates and extents. Group averages and standard error across participants are shown, along with predictions (colored solid lines) from the PEA model. (**B**) The adaptation extent (cycle 100–110) exhibits a nonlinear dependency on perturbation size, conforming to a concave function as prescribed by PEA (purple line). Color dots and error bars denote the mean and standard error across participants in each group. (**C–D**) The same data fitted with the PReMo model and the causal inference model. For more details, refer to *Figure 3—figure supplement 1*.

The online version of this article includes the following figure supplement(s) for figure 3:

**Figure supplement 1.** Model fitting for observed implicit adaptation in Experiment 2.

**Figure supplement 2.** Correlation between initial learning rate and adaptation extent in Experiment 2.

**Figure supplement 3.** Model fitting of single-trial learning from Experiment 2 of *Tsay et al., 2021a*.

**Figure supplement 4.** Alternative Model fitting with PEA for Experiment 2; see details in Appendix 2.

noticed that our adaptation phase might not be long enough for small clamp-size conditions to fully adapt (e.g. 2°, 4°, *Figure 3A*). Nevertheless, we operationally used the adaptation level achieved at the end of the adaptation phase as the measure of adaptation extent. Critically, the adaptation extent displayed a concave pattern: increasing steeply for smaller perturbations and tapering off for larger ones (*Figure 3B*). A one-way ANOVA revealed a significant group difference in adaptation extent ($F_{(6,83)} = 12.108$, p=1.543e-09). Planned contrasts indicated that 8°, 16°, and 32° perturbations

did not differ from each other (all p>0.417, with Tukey-Kramer correction), consistent with earlier evidence of invariant implicit adaptation (*Kim et al., 2018*). However, 64° and 95° perturbations led to significantly reduced adaptation extents compared to 8° (p=3.194e-05 and 5.509e-06, respectively), supporting the concave pattern as a more accurate portrayal of implicit adaptation across varying perturbation sizes.

Importantly, the PEA model, when augmented with visual uncertainty data from Experiment 1, precisely predicts this size-dependent adaptation behavior (*Figure 3B*). Beyond adaptation extent, the model also accurately predicts the trial-by-trial adaptation across all seven participant groups, employing a single parameter set (R$^2$=0.975; *Figure 3A*). The model had only four free parameters (A=0.974, B=0.208, $\sigma_p$ = 11.119°, $\sigma_u$ = 5.048°; *Supplementary file 1a*). Remarkably, both the retention rate A and learning rate B are consistent with previous studies focusing on visuomotor rotation adaptation (*Albert et al., 2022*). We also quantified proprioceptive uncertainty ($\sigma_p$) in a subset of participants (n=13) using a similar 2AFC procedure as in Experiment 1. We found that $\sigma_p$ was 9.737°±5.598° (*Figure 4—figure supplement 1*), which did not statistically differ from the $\sigma_p$ value obtained from the model fitting (two-tailed t-test, p=0.391). We also quantified $\sigma_u$ as the standard deviation of movement direction from baseline in Experiment 2. The calculated standard deviation is 5.128°±0.190°, also not significantly differ from the estimated $\sigma_u$ by model fitting (two-tailed t-test, p=0.693). To further verify the model, a separate data fitting without assuming a linear function in the visual uncertainty was conducted on the data of Experiment 2 (see details in Appendix 2 and *Figure 3—figure supplement 4*). The estimated visual uncertainty has a strong linear relationship with perturbation size (R=0.991, p<0.001). In addition, the slope of model-fitted visual uncertainty is very close to the values we obtained in Experiment 1. In summary, the perceptual parameters obtained in Experiment 1, when incorporated into the PEA model, effectively explain the implicit adaptation behaviors observed in different participant groups in Experiment 2.

In comparative analysis, the PReMo model yields a substantially lower R$^2$ value of 0.749 (*Figure 3—figure supplement 1B*). It tends to underestimate adaptation for medium-size perturbations and overestimate it for large ones (*Figure 3C*; see also *Figure 3—figure supplement 1B* for trial-by-trial fitting). Another alternative is the causal inference model, previously shown to account for nonlinearity in motor learning (*Mikulasch et al., 2022*; *Wei and Körding, 2009*). Although this model has been suggested for implicit adaptation (*Tsay et al., 2021a*), it fails to reproduce the observed concave adaptation pattern (*Figure 3D*, *Figure 3—figure supplements 1–3C*). The model aligns well with adaptations to medium-size perturbations (8°, 16°, and 32°) but falls short for small and large ones, yielding an R$^2$ value of 0.711 (see *Figure 3—figure supplement 1C* for trial-by-trial fits). Model comparison metrics strongly favor the PEA model over both the PReMo and causal inference models, as evidenced by AIC scores of 2255, 3543, and 3283 for the PEA, PReMo, and causal inference models, respectively (*Supplementary file 1b*). In summary, it is the eccentricity-induced visual uncertainty that most accurately accounts for the implicit adaptation profile across a broad spectrum of perturbation sizes, rather than saturated visual influence or causal inference of error.

## Experiment 3: Cue combination accounts for changes in proprioception measures during implicit adaptation

Motor adaptation not only recalibrates the motor system but also alters proprioception (*Rossi et al., 2021*) and even vision (*Simani et al., 2007*). For implicit adaptation, the perceived hand location shifted towards the clamped cursor immediately upon the introduction of the error-clamp, but then gradually drifted away from the clamped cursor (*Tsay et al., 2020*), shown as a gradual change from negative to positive relative to the target (*Figure 4A*). This perceived hand location was verbally reported after an active movement with the error-clamp, with the hand staying at the end of the reach. Thus, it is not the so-called proprioceptive recalibration that is typically probed using a passively located hand (*Cressman and Henriques, 2009*; *Mostafa et al., 2019*; *'t Hart and Henriques, 2016*; *Tsay et al., 2021a*). Nevertheless, this intriguing gradual drift of active localization of the hand is informative of the underlying mechanism of implicit adaptation. The PReMo model proposes that the initial negative drift reflects a misperceived hand location, which gradually reduces to zero, and the late positive drift reflects the influence of visual calibration of the target (*Tsay et al., 2022c*). However, this assumption lacks empirical validation, and how visual recalibration relates to a proprioceptive measurement without visual cues is not laid out either. In contrast, we suggest that the perceived hand

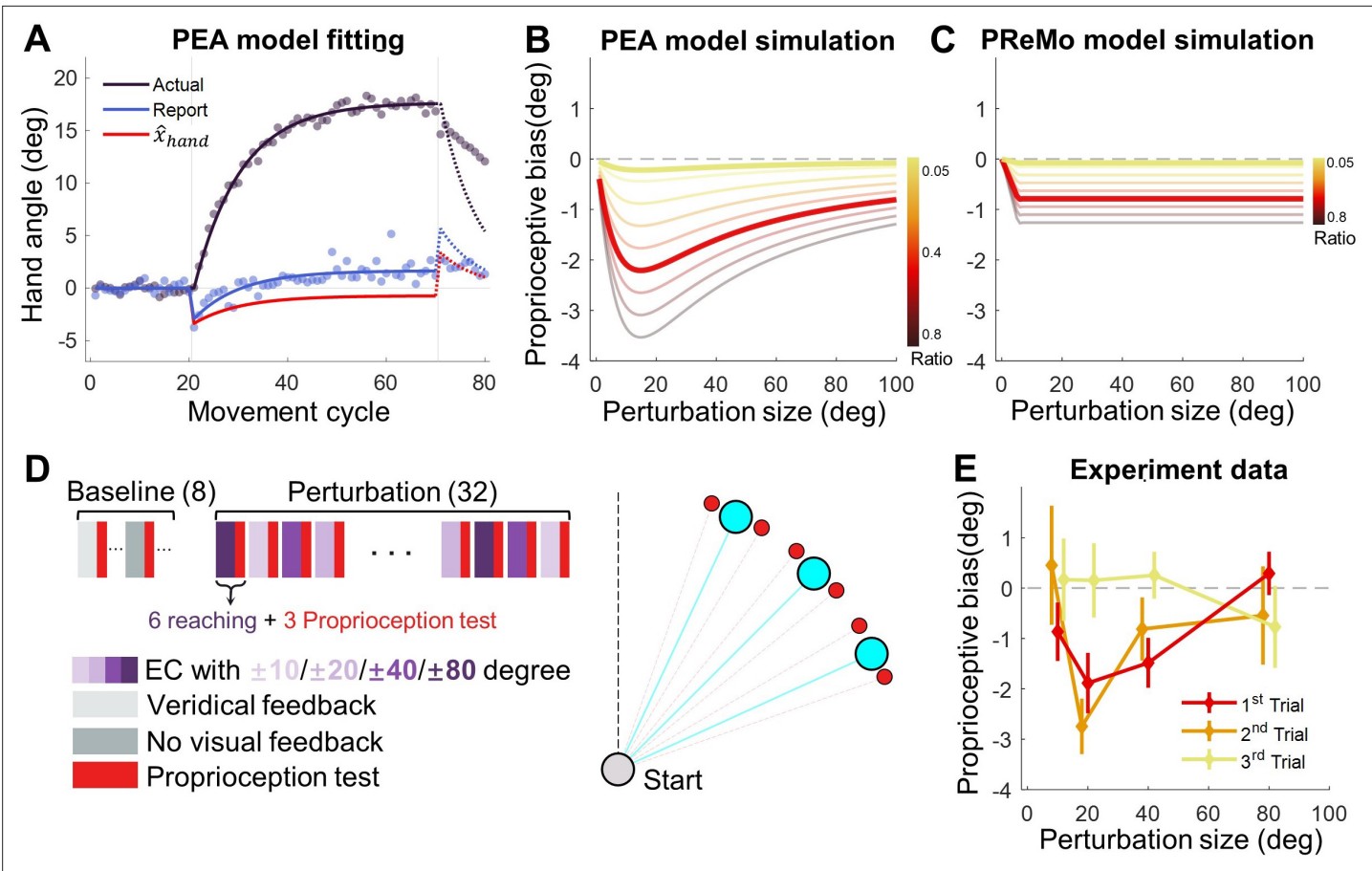

**Figure 4.** Proprioceptive data fitting and results from Experiment 3. (**A**) The data from *Tsay et al., 2020* are presented alongside the fitting of the PEA model. Participants adapting to a 30° error-clamp perturbation were required to report the location of their adapted hand using visual aids of numbers. The report was provided when the hand stayed at the end of movement. Initially, the proprioceptive estimate of the hand is biased toward the visual cursor (negative in the plot) and then gradually shifts toward the hand (positive in the plot). This trend is accurately captured by the PEA model: lines represent model fitting results, with the adapted hand direction in indigo and the reported hand direction in blue. The hand direction estimate ($\hat{x}_{Hand}$, *Equation 1*) following a reach movement is shown in red. (**B–C**) Model simulations for proprioceptive bias from the PEA and PReMo models. Color gradients denote the simulations with varying ratio between the weights of $\hat{x}_{Hand}$ and $x_p$, the two cues available for estimating the hand direction. Note that the two models prescribe distinct profiles for the dependency of proprioception bias on perturbation size. (**D**) Experimental design. A reaching block, either with or without visual perturbations, is followed by a proprioception test block. The size and direction of the visual perturbation vary across blocks. The proprioception test is conducted when the hand is passively moved to a target (red dots) situated near the reaching target (blue dots). (**E**) The observed proprioceptive bias as a function of perturbation size (n = 11). Data from the three proprioception test trials are separately plotted. The first trial reveals proprioception biases that form a concave function of perturbation size.

The online version of this article includes the following figure supplement(s) for figure 4:

**Figure supplement 1.** Proprioception uncertainty estimation results.

location is based on the same Bayesian cue combination principle we laid out in the PEA model. In this particular setting, the perceived hand location at the end of each reach is determined by the proprioceptive cue ($x_p$) and the estimated hand position under the influence of clamped feedback ($\hat{x}_{Hand}$, *Equation 1*), the two relevant cues at the time of reporting relative to the target position.

During early adaptation, $\hat{x}_{Hand}$ is biased towards the clamped feedback (small negative value, *Figure 4A*), while $x_p$ remains near the target as the motor system has yet to adapt (close to 0, *Figure 4A*). This results in an initial negative proprioceptive bias. As adaptation progresses, although $\hat{x}_{Hand}$ remains biased with a small negative value, $x_p$ gradually shifts in the positive direction due to adaptation (value changed from near zero to increasingly positive), resulting in an increasingly positive perceived hand location. Remarkably, the PEA model can predict these temporal changes in the localization of an actively located hand with high accuracy ($R^2$=0.982; *Figure 4A*).

We postulate that the same Bayesian cue combination governs not only the active localization of the hand in *Figure 4A* but also the proprioceptive recalibration that is probed by the passive localization of the hand. With the biased $\hat{x}_{Hand}$ during adaptation, our PEA model can make specific quantitative predictions about the relationship between proprioception changes, measured by a passively located hand, and visual perturbation size. While traditional visuomotor paradigms suggest either invariant (*Modchalingam et al., 2019*) or linear increases in proprioceptive recalibration with an increase in visual-proprioceptive discrepancy (*Salomonczyk et al., 2011*), the PEA model prescribes a concave function in relation to visual perturbation size simply due to the fact that $\hat{x}_{Hand}$ follows a concave function in relation to the perturbation size (*Figure 4B*, *Equation 9*).

To empirically test this prediction, Experiment 3 (n=11) measured participants' proprioceptive recalibration during implicit adaptation using a procedure similar to the error-clamp perturbations in Experiment 2. After each block of six adaptation trials, participants' right hands were passively moved by a robotic manipulandum, and they indicated the perceived direction of their right hand using a visually represented 'dial' controlled by their left hand (Figure 7B). This method quantifies proprioceptive recalibration during adaptation (*Cressman and Henriques, 2009*). Each adaptation block was followed by three such proprioception test trials. The alternating design between adaptation and proprioception test blocks allowed us to assess proprioceptive biases across varying perturbation sizes, which consisted of ± 10°, ± 20°, ± 40°, and ± 80°, to cover a wide range (*Figure 4D*).

Our findings confirmed a typical proprioceptive recalibration effect, as the perceived hand direction was biased towards the visual perturbation (*Figure 4E*). Importantly, the bias in the initial proprioception test trial exhibited a concave function of perturbation size. A one-way repeated-measures ANOVA revealed a significant effect of perturbation size (F(3,30)=3.603, p=0.036), with the 20° and 40° conditions displaying significantly greater proprioceptive bias compared to the 80° condition (pairwise comparisons: 20° v.s. 80°, p=0.034; 40° v.s. 80°, p=0.003). The bias was significantly negative for 20° and 40° conditions (p=0.005 and p=0.007, respectively, with one-tailed t-test) but not for 10° and 80° conditions (p=0.083 and p=0.742, respectively). The concave pattern aligns well with the PEA model's predictions (*Figure 4B*), further consolidating its explanatory power.

This stands in contrast to the PReMo model, which assumes a saturation for the influence of the visual cue on the hand estimate (*Equations 12 and 13*). As a result, PReMo's predicted proprioceptive bias follows a ramp function, deviating substantially from our empirical findings (*Figure 4C*). The causal inference model, which mainly focuses on the role of visual feedback in error correction, cannot directly predict changes in proprioceptive recalibration.

Interestingly, we observed that the proprioceptive bias was reduced to insignificance by the third trial in each proprioception test block (one-tailed t-test, all p>0.18; *Figure 4E*, yellow line). The observed proprioceptive bias is formally modeled as a result of the biasing effect of the perceived hand estimate $\hat{x}_{Hand}$. In our mini-block of passive localization, the participants neither actively moved nor received any cursor perturbations for three trials in a row. Thus, the fact that the measured proprioceptive bias is reduced to nearly zero at the third trial suggests that the effect of perceived hand estimate $x_{Hand}$ decays rather rapidly.

## Experiment 4: Differential impact of upregulated visual uncertainty on implicit adaptation across perturbation sizes

Thus far, we have presented both empirical and computational evidence underscoring the pivotal role of perceptual error and visual uncertainty in implicit adaptation. It is crucial to note, however, that this evidence is arguably correlational, arising from natural variations in visual uncertainty as a function of perturbation size. To transition from correlation to causation, Experiment 4 (n=19) sought to directly manipulate visual uncertainty by blurring the cursor, thereby offering causal support for the role of multimodal perceptual error in implicit adaptation.

By increasing visual uncertainty via cursor blurring, we hypothesized a corresponding decrease in adaptation across all perturbation sizes. Notably, the PEA model predicts a size-dependent attenuation in adaptation: the reduction is less marked for smaller perturbations and more pronounced for larger ones (*Figure 5A*). This prediction diverges significantly from those of competing models. The PReMo model, operating under the assumption of a saturation effect for large visual perturbations, predicts that cursor blurring will only influence adaptation to smaller perturbations, leaving adaptation to larger perturbations unaffected (*Figure 5B*). The causal inference model makes an even more

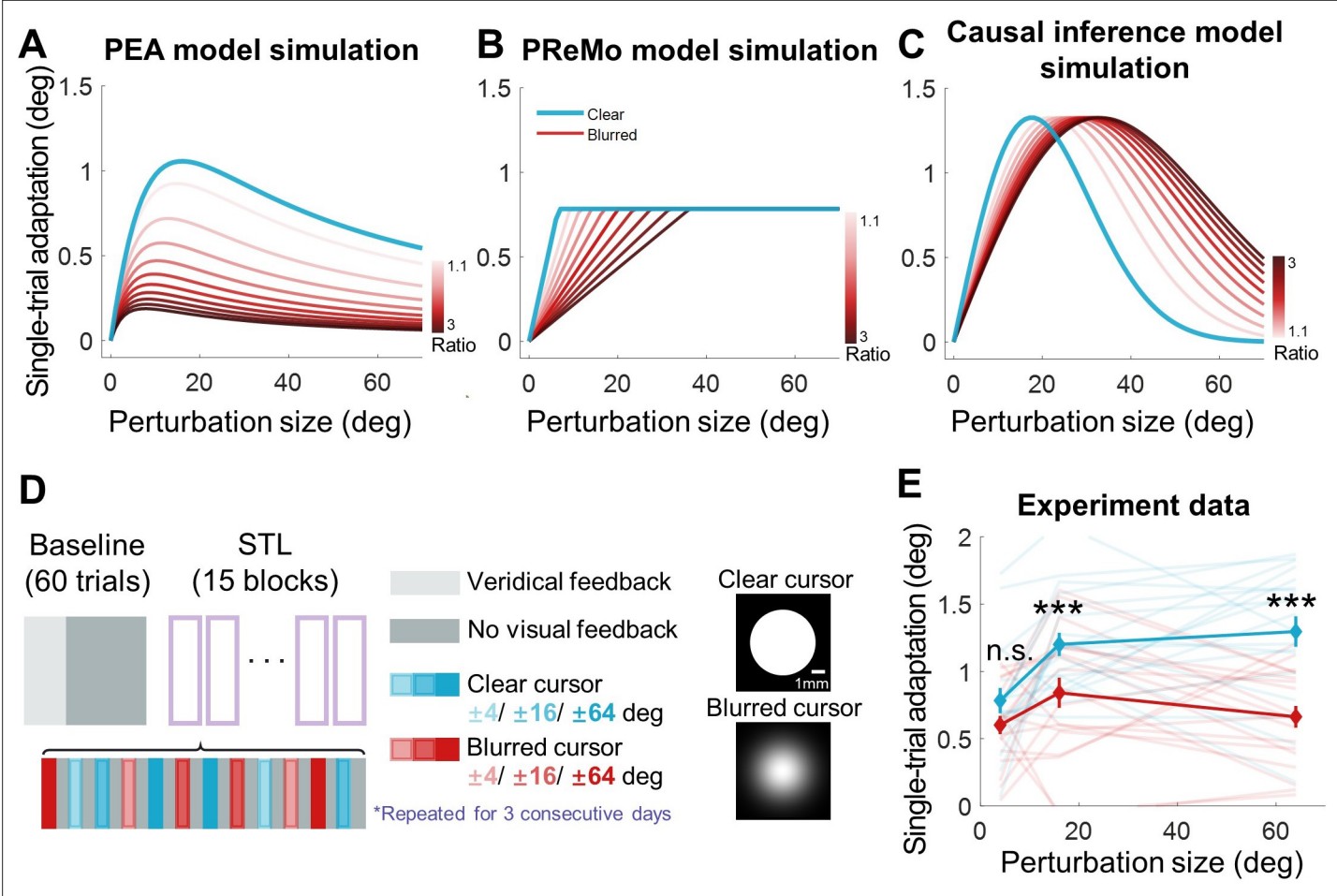

**Figure 5.** Results of Experiment 4. (**A–C**) Model simulations for single-trial learning under different visual uncertainty levels, shown separately for the PEA, PReMo and causal inference models. Blue curves represent simulated learning based on model parameters estimated from Experiment 2. Curves with red gradient indicate simulations with increasing levels of visual uncertainty, color coded by the ratio of visual uncertainty for the blurred cursor to that of the clear cursor. (**D**) Experimental design. Following 60 baseline trials without perturbations, participants completed 15 mini-blocks of error-clamp adaptation over three successive days. Each mini-block features 12 different types of error-clamp perturbations, distinguished by two cursor presentations (blurred or clear cursor) and six clamp sizes. Each perturbation trial, varied randomly in perturbation type, is flanked by two no-feedback trials. The change in hand direction between these two no-feedback trials serves to quantify singe-trial learning. (**E**) The single-trial learning with the blurred cursor is less than that with the clear cursor, but the difference is non-monotonic across perturbation size (n = 19, *** denote p<0.001).

nuanced prediction: it anticipates that the blurring will lead to a substantial reduction in adaptation for small perturbations, a diminishing effect for medium perturbations, and a potential reversal for large perturbations (***Figure 5C***). This prediction results from the model's core concept that causal attribution of the cursor to self-action—which directly dictates the magnitude of adaptation—decreases for small perturbations but increases for large ones when overall visual uncertainty is elevated.

Starting from the above predictions, Experiment 4 was designed to assess the impact of elevated visual uncertainty across small (4°), medium (16°), and large (64°) perturbation sizes. Visual uncertainty was augmented by superimposing a Gaussian blurring mask on the cursor (***Burge et al., 2008***). Each participant performed reaching tasks with either a standard or blurred clamped cursor for a single trial, bracketed by two null trials devoid of cursor feedback (***Figure 5D***). These three-trial mini-blocks permitted the quantification of one-trial learning as the directional difference of movements between the two null trials. To preclude the cumulative effect of adaptation, perturbation sizes and directions were randomized across mini-blocks.

Crucially, our findings corroborated the predictions of the PEA model: visual uncertainty significantly diminished adaptation for medium and large perturbations (16° and 64°), while leaving adaptation for small perturbations (4°) largely unaffected (***Figure 5E***). A two-way repeated-measures

ANOVA, with two levels of uncertainty and three levels of perturbation size, revealed a significant main effect of increased visual uncertainty in reducing implicit adaptation ($F(1,18) = 42.255$, $p=4.112e-06$). Furthermore, this effect interacted with perturbation size ($F(2,36) = 5.391$, $p=0.012$). Post-hoc analyses demonstrated that elevated visual uncertainty significantly attenuated adaptation for large perturbations ($p=2.877e-04$, $d=0.804$ for 16°; $p=1.810e-05$, $d=1.442$ for 64°) but exerted no such effect on small perturbations ($p=0.108$, $d=0.500$). These empirical outcomes are not congruent with the predictions of either the PReMo or the causal inference models (*Figure 5B and C*). This lends compelling empirical support to the primacy of perceptual error in driving implicit adaptation, as posited by our PEA model.

## Discussion

In this study, we elucidate the central role of perceptual error, derived from multimodal sensorimotor cue integration, in governing implicit motor adaptation. Utilizing the classical error-clamp paradigm, we uncover that the overcompensation observed in response to small perturbations arises from a sustained perceptual error related to hand localization, and the saturation effect commonly reported in implicit adaptation is not an intrinsic characteristic of cerebellum-based learning but is attributable to increasing sensory uncertainty with increasing visual perturbation eccentricity—a factor hitherto neglected in existing models of sensorimotor adaptation. Contrary to conventional theories that describe implicit adaptation as either saturated or invariant (*Kim et al., 2018*; *Tsay et al., 2022c*), our data reveal a concave dependency of implicit adaptation on visual perturbation size, characterized by diminishing adaptation in response to larger perturbations. Notably, our Perceptual Error Adaptation (PEA) model, calibrated using perceptual parameters from one set of participants, provides a robust account of implicit adaptation in separate groups subjected to varying perturbations. The model further successfully captures the perceptual consequences of implicit adaptation, including the continuous shifts in hand localization after active movements during adaptation process (*Tsay et al., 2020*) and the proprioceptive recalibration probed by localization of the passively moved hand. Lastly, we manipulated visual uncertainty independently of perturbation size and demonstrated that upregulating visual uncertainty selectively attenuated adaptation in the context of larger perturbations while leaving smaller perturbations unaffected. These empirical results, inconsistent with predictions from existing models, underscore the conceptual and quantitative superiority of our PEA model. Our study advocates for a revised understanding of implicit motor adaptation; that is, it is driven not by sensory prediction errors but by perceptual errors in localizing one's effector, conforming to Bayesian cue combination.

The importance of perceptual error was first implied in the PReMo model (*Tsay et al., 2022b*), which also proposed that a misperceived hand location is the driving signal for implicit adaptation. However, this previous work improperly called this perceptual error a proprioceptive error, leading to the impression that it is a type of sensory error (but see their new unpublished *Tsay et al., 2024*). The two models differ fundamentally in their conceptualization of how different cues contribute to the error signal. The PReMo model posits two intermediate perceptual variables with Bayesian cue integration: a visual estimate of the cursor and a proprioceptive estimate of the hand (*Tsay et al., 2022c*). The final error signal for adaptation is presumed to be a proprioceptive error, not from further Bayesian cue combination, but from a visual-to-proprioceptive bias that is governed by a predetermined, ramp-like visual influence that saturates around a 6–7° visual-proprioceptive discrepancy (*Equation 13*). These assumptions lack empirical validation. In fact, our findings in Experiment 3 indicate that proprioceptive recalibration follows a concave function with respect to visual perturbation size, contradicting the ramp-like function assumed by PReMo. Moreover, the presupposed ramp-like visual influence generates a rigid prediction for a ramp-like adaptation extent profile, which is at odds with the concave adaptation pattern observed in Experiment 2 and in a similar study involving trial-by-trial learning (*Tsay et al., 2021a*). Furthermore, the PReMo model predicts that upregulating visual uncertainty will selectively reduce adaptation to small perturbations while sparing large ones. This is inconsistent with our findings in Experiment 4, which demonstrated that upregulating visual uncertainty substantially impacted adaptation more to larger perturbations than small ones. Lastly, though PReMo has the potential to explain the temporal shifts in perceived hand location during adaptation, the authors resorted to separate mechanisms of proprioceptive and visual recalibration at different phases of adaptation to explain these shifts (*Tsay et al., 2020*). In summary, the PReMo model's

assumptions introduce limitations that make it less consistent with empirical observations, particularly concerning the nonlinearities observed in both motoric and perceptual aspects of implicit adaptation.

The causal inference framework (*Wei and Körding, 2009*) fails to predict sensorimotor changes in implicit adaptation adequately. For instance, it underestimates the adaptation extent for large perturbations and incorrectly predicts that increasing visual uncertainty would augment, rather than reduce, adaptation to large perturbations. The casual inference model is also based on Bayesian principle, then why does it fail to account for the implicit adaptation? We postulate that the failure of the causal inference model is due to its neglect of visual uncertainty as a function of perturbation size, as we revealed in Experiment 1. In fact, previous studies that advocating the Bayesian principle in motor adaptation have largely focused on experimentally manipulating sensory cue uncertainty to observe its effects on adaptation (*Burge et al., 2008*; *He et al., 2016*; *Körding and Wolpert, 2004*; *Wei and Körding, 2010*), similar to our Experiment 4. Our findings suggest that causal inference of perturbation alone, without incorporating visual uncertainty, cannot fully account for the diverse findings in implicit adaptation. The increase in visual uncertainty by perturbation size is substantial: our Experiment 1 yielded an approximate sevenfold increase from a 4° perturbation to a 64° perturbation. We have attributed this to the fact that people fixate in the desired movement direction during movements. Interestingly, even for conventional visuomotor rotation paradigm where people are required to 'control' the perturbed cursor, their fixation is also on the desired direction, not on the cursor itself (*de Brouwer et al., 2018a*; *de Brouwer et al., 2018b*). Thus, we postulate that a similar hike in visual uncertainty in other 'free-viewing' perturbation paradigms. Future studies are warranted to extend our PEA model to account for implicit adaptation in other perturbation paradigms.

Our research contributes to an ongoing debate concerning the driving forces behind error-based motor learning, specifically addressing the question of whether implicit adaptation is driven by target error or sensory prediction error (*Albert et al., 2022*; *Izawa and Shadmehr, 2011*; *Leow et al., 2020*; *Mazzoni and Krakauer, 2006*; *McDougle et al., 2015*; *Miyamoto et al., 2020*; *Taylor and Ivry, 2011*; *Tseng et al., 2007*). Most empirical data fueling this debate stem from traditional motor adaptation paradigms where explicit and implicit learning co-occur and interact. In these paradigms—visuomotor rotation being a prime example—target error (TE) is defined as the disparity between the target and the perturbed cursor, while sensory prediction error (SPE) is the disparity between the predicted and actual cursor. These two types of error, sensory (specifically, visual) in nature, differ simply due to the presence of explicit learning: the predicted (desired and aiming) direction deviates from the target direction when explicit learning is present (*Taylor et al., 2014*). Our study suggests that neither target error nor sensory (visual) prediction error drives implicit adaptation in the error-clamp paradigm. The error-clamp paradigm isolates implicit learning and thus eliminates potential confounds from explicit learning. In this paradigm, the predicted and target directions are aligned, and the target error and sensory prediction error effectively refer to the same visual discrepancy. However, either TE or SPE, if plugged in the classical state-space models, is able to account for the nuanced features of implicit adaptation (*Tsay et al., 2022c*). In contrast, our PEA model reframes the perturbing cursor as a visual cue influencing the perceptual estimation of hand location rather than as a source of visual error. The resultant bias in hand estimation from the desired target serves as the actual error signal. This leads us to posit that the error signal driving implicit sensorimotor adaptation is fundamentally perceptual rather than sensory. From a normative standpoint, this perceptual error could be construed either as a predictive or performance error (*Albert et al., 2022*), but importantly, it is not tied to a specific modality (i.e. vision or proprioception). Instead, it directly pertains to the perceptual estimate that is crucial for task execution, that is, bringing the hand in the desired direction.

The concept of perceptual error-driven learning can be extrapolated to various motor learning paradigms, including other motor adaptation tasks. For instance, in visuomotor rotation tasks, explicit learning manifests as a deviation in the aiming direction from the visual target, whereas implicit learning manifests as a further deviation the actual hand position from this aiming direction (*Taylor et al., 2014*). With the presence of re-aiming, the perturbed cursor still deviates from the re-aiming direction and thus produces the perceptual bias of the hand from the desired direction, which subsequently drives implicit adaptation. In this scenario, the perceptual error is defined as the difference between the perceptual estimate of the hand and the 're-aiming' direction, which serves as the new 'target' when explicit learning is in play. Our PEA model would predict similar saturation effects in implicit adaptation for this conventional adaptation paradigm, comparable to for the error-clamp

paradigm. Indeed, existing findings from the conventional adaptation support this prediction: the implicit adaptation follows either a saturation effect (*Bond and Taylor, 2015*; *Neville and Cressman, 2018*) or a concave pattern (*Tsay et al., 2022a*) across a range of perturbation sizes. Furthermore, according to the PEA framework, this perceptual error is anchored on the aiming target, thereby naturally predicting that implicit and explicit adaptations should interact in a complementary manner, a notion that aligns with recent theories on their interaction (*Albert et al., 2022*; *Miyamoto et al., 2020*).

Besides visuomotor rotation, all the primary motor adaptation paradigms, including prism adaptation (e.g. *Petitet et al., 2018*; *von Helmholtz, 1867*), visuomotor gain adaptation (e.g. *Pearson et al., 2010*), and force field adaptation (e.g. *Shadmehr and Mussa-Ivaldi, 1994*), involves locating one's effector under the influence of perturbed sensory feedback. Thus, we postulate that perceptual error elicited by sensory perturbation might play a similar significant role in driving implicit learning in these diverse task paradigms. It is noteworthy that error-based learning also plays a major role in motor skill learning, including de novo skill learning and motor acuity learning (*Krakauer et al., 2019*). Unlike motor adaptation, motor skill learning currently lacks computational models to account for its error-based learning component, owing to the complexity and diversity of skill learning tasks. We believe that Bayesian cue combination and perceptual error about locating one's effect might serve as a starting point for theorization of the implicit processes in skill learning.

Our study also provides a new perspective on explaining proprioceptive changes during motor adaptation, advocating for a Bayesian cue combination framework. Previously, the change in proprioceptive hand localization during motor adaptation has been ascribed to visual-proprioceptive discrepancy-induced recalibration (*Ruttle et al., 2018*; *Salomonczyk et al., 2013*) and/or altered sensory prediction caused by the adapted forward internal model (*Mostafa et al., 2019*; *'t Hart and Henriques, 2016*). To dissect these components, researchers have often compared proprioceptive localization in actively moved (*Ruttle et al., 2021*; *'t Hart and Henriques, 2016*) versus passively placed (passive localization, e.g. Experiment 3) hands during adaptation, attributing the smaller bias in passive localization to recalibration alone. The difference between the two is then considered to reflect altered sensory prediction due to motor adaptation (*Mostafa et al., 2019*; *Rossi et al., 2021*). But these conceptual divisions lack computational models for validation. For instance, researchers have shown that proprioceptive recalibration in visuomotor adaptation is either a fixed proportion (e.g. 20%) of the visual-proprioceptive discrepancy (*Henriques and Cressman, 2012*; *Ruttle et al., 2021*) or largely invariant (*Modchalingam et al., 2019*). However, a computational model has yet to be proposed elucidate the underlying mechanism for these diverse findings. In fact, cross-sensory calibration typically follows the Bayesian principle, as shown in task paradigms other than motor adaptation (*Stetson et al., 2006*; *Wozny and Shams, 2011*). We propose that proprioceptive changes elicited by motor adaptation, no matter measured by a passive or active localization method, are governed by the same Bayesian cue combination principle and the same critical cue, that is (mis) perceived hand location ($\hat{x}_{Hand}$). Given that $\hat{x}_{Hand}$ has a concave dependency to visual perturbation size, we would observe a corresponding concave proprioceptive recalibration profile.

Besides highlighting the importance of Bayesian cue combination, our study also suggests that proprioceptive measurements should be understood by considering available cues in the specific task setting. For both localization tests, that is the passive (Experiment 3) and active localization (*Tsay et al., 2020*; *Figure 4A*), the hand stayed at the trial end when the participants reported their perceived hand location. Thus, the reported hand location is determined by the just-experienced $\hat{x}_{Hand}$ and the actual proprioceptive cue. In the case of active localization, the proprioceptive cue becomes increasingly large (biased to the positive direction in reference to the target), driven by the adaptation process. In this sense, active localization indeed serves as a multifaceted reflection of both the internal model (the adapted hand, supplying $x_p$) and proprioceptive recalibration (the perceived hand after a movement, $\hat{x}_{Hand}$), as proposed by previous researchers (*Mostafa et al., 2019*; *Rossi et al., 2021*). During the initial stages of perturbation, the immediate negative bias in active localization is predominantly attributable to rapid proprioceptive recalibration. This is evidenced by a sudden shift in the estimated hand position ($\hat{x}_{Hand}$ ; *Figure 4A*), occurring before the internal model has had sufficient time to adapt. Then, why does active localization in traditional motor adaptation paradigms yield a largely stable bias (*Ruttle et al., 2016*; *Ruttle et al., 2021*)? We note that traditional visuomotor rotation paradigms invokes a rapid initial explicit learning, driving the adaptation to its

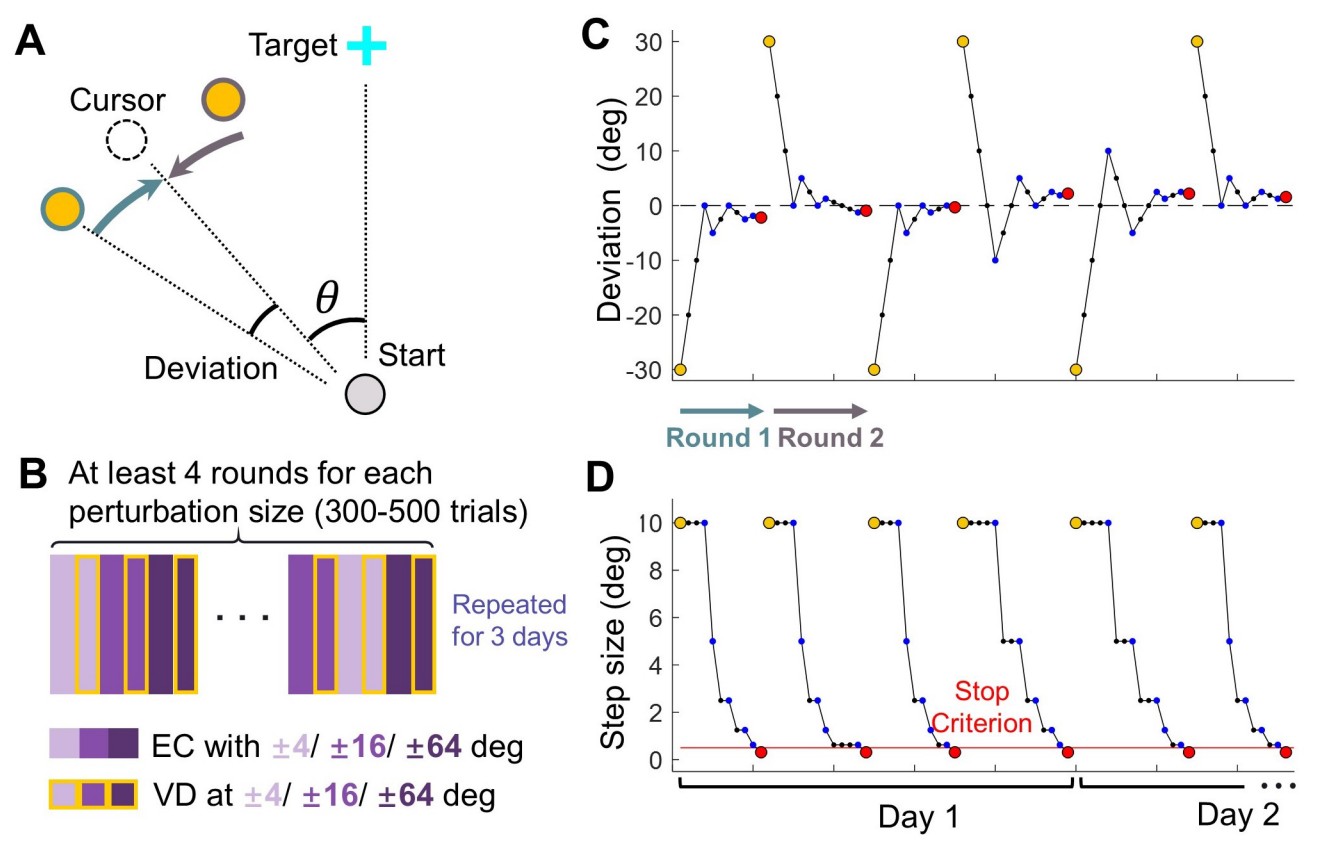

**Figure 6.** Design of Experiment 1. (**A**) Top-down view of the setup in visual discrimination task. The reference point (yellow) was presented either CW or CCW relative to the clamped cursor (dashed circle), which has a perturbation size $\theta$ . (**B**) Trial structure of the visual discrimination task. Purple rectangles represent error-clamped trials with varying perturbation size, rectangles with yellow edges represent the ensuing visual discrimination test for each perturbation size. (**C–D**) Exemplary sequences of the reference point: These sequences illustrate the deviation of the reference point from the cursor (**C**) and the changing step size across trials (**D**), following the PEST algorithm. Individual trials are represented by blue dots. Yellow and red dots mark the initiation and termination of each round of trials, respectively. In each round, the reference point starts on either the CW or CCW side of the cursor; In the subsequent round, it starts on the opposite side.

asymptote quickly. At the same time, previous investigations have predominantly measured active localization when adaptation has asymptoted (*Henriques and Cressman, 2012*; *Modchalingam et al., 2019*; *Mostafa et al., 2019*; *Salomonczyk et al., 2011*; *Salomonczyk et al., 2013*; *Tsay et al., 2021b*). Consequently, these studies overlook the evolving effect of the adaptation. In contrast, the gradual nature of implicit adaptation, shown in the error-clamp paradigm here, provides a unique opportunity to uncover the underlying mechanisms governing changes in active localization during the adaptation process.

Notably, our model aligns with previous findings that show a positive correlation between proprioceptive recalibration, measured by the passive localization method, and motor adaptation based on individual differences (*Ruttle et al., 2021*; *Salomonczyk et al., 2013*; *Tsay et al., 2021b*). Unlike existing theories that posit proprioceptive recalibration either as a component (*Modchalingam et al., 2019*; *Mostafa et al., 2019*; *Ruttle et al., 2021*) or a driver for implicit adaptation (*Tsay et al., 2022c*), our PEA model provides a mechanistic and empirically testable framework. It posits that the misestimation of hand position ($\hat{x}_{Hand}$)—induced by the recent perturbation—serves as the driving factor for both implicit adaptation and changes in proprioception. In other words, this misestimation is the common cause for both implicit adaptation and proprioceptive recalibration. This is why the concave dependency of $\hat{x}_{Hand}$ on perturbation leads to similar concave perturbation dependencies in both implicit adaptation and proprioceptive recalibration. Updated on a trial-by-trial basis, this misestimation exerts immediate effects, manifesting as an abrupt negative bias (*Figure 4A*). Additionally, its influence decays rapidly, becoming negligible within three trials (*Figure 6C*). These converging lines

of evidence strongly suggest that perceptual misestimation of hand position is central to the process of proprioceptive recalibration during adaptation.

Our findings contribute nuanced perspectives to the modulation of implicit learning rate by factors beyond visual perturbation size. Previous studies have shown that environmental inconsistency—defined as the inconsistency of visual errors—reduced the rate (*Herzfeld et al., 2014*; *Hutter and Taylor, 2018*) or extent (*Albert et al., 2021*) of implicit adaptation. Baseline motor variance in unperturbed conditions has been shown to increase implicit adaptation rate, proposed as a sign of better exploratory learning (*Wu et al., 2014*). These studies interpret such phenomena as parametric changes in the learning rate in relation to visual errors, conceptualized as alterations to the *B* parameter in existing models. However, an apparent change in learning rate due to visual errors does not necessarily signify parametric modification, but may attribute to other factors that influence the use of visual cues (*He et al., 2016*), such as visual uncertainty in our case. Previous research has also pointed to various alternative factors for determining apparent learning rate, including error discounting based on causal inference of error (*Wei and Körding, 2009*), proprioceptive uncertainty (*Ruttle et al., 2021*; *Tsay et al., 2021b*), and state estimation uncertainty (*He et al., 2016*). In line with these previous studies, our work suggests a shift in perspective: the driving error signal for implicit learning should be considered as perceptual, rather than merely visual. Hence, all the aforementioned factors can be considered as contributing to the perceptual estimate. This paradigmatic shift could serve as a cornerstone for future research aimed at understanding how learning rates change under varying conditions.

Our new framework opens avenues for exploring the memory characteristics of implicit learning. Traditional motor adaptation often exhibits 'savings', or accelerated relearning upon re-exposure to a perturbation (*Della-Maggiore and McIntosh, 2005*; *Huberdeau et al., 2019*; *Krakauer et al., 2005*; *Landi et al., 2011*). In contrast, implicit adaptation has been found to exhibit a decreased learning rate during re-adaptation (*Avraham et al., 2021*), a phenomenon attributed to conditioning (*Avraham et al., 2021*) or associative learning mechanisms (*Avraham et al., 2022*). Investigating this 'anti-saving' effect will yield insights into the unique memory properties of implicit learning. Although our current PEA model is structured around single-epoch learning and does not directly address this question, it does raise new, testable hypotheses. For example, is the reduced adaptation rate during relearning attributable to a down-weighting of perturbed visual feedback in cue combination, or does it reflect a parametric alteration in the learning rate? Another noteworthy aspect of implicit learning is its remarkably slow decay rate. It has been observed that the number of trials required to wash out the implicit adaptation exceeds the number of trials needed to establish it (*Avraham et al., 2021*; *Tsay et al., 2020*). In the context of our perceptual error framework, this raises the possibility that washout phases might be governed by state updating involving a distinct set of sensorimotor cues or an alternative updating mechanism, such as memory formation and selection (*Oh and Schweighofer, 2019*).

## Methods
### Participants
We recruited 115 college students from Peking University (77 females, 38 males, 22.05±2.82 years, mean ± SD). Participants were all right-handed according to the Edinburgh handedness inventory (*Oldfield, 1971*) and had normal or corrected-to-normal vision. Participants were naïve to the purpose of the experiment and provided written informed consent, which was approved by the Institutional Review Board of the School of Psychological and Cognitive Sciences, Peking University. Participants received monetary compensation upon completion of the experiment.

### Apparatus
In Experiment 1, 2, and 4, participants were seated in front of a vertically-placed LCD screen (29.6x52.7 cm, Dell, Round Rock, TX, US). They performed the movement task with their right hand, holding a stylus and slide it on a horizontally placed digitizing tablet (48.8x30.5 cm, Intuos 4 PTK-1240, Wacom, Saitama, Japan). In Experiment 1, a keyboard was provided to the participants' left hand to enable them to report the direction of visual stimuli in the discrimination task. A customized wooden shelter was placed above the tablet to block the peripheral vision of the right arm. In Experiment 1 and 4, participants placed their chin on a chin rest attached to the wooden shelter to stabilize

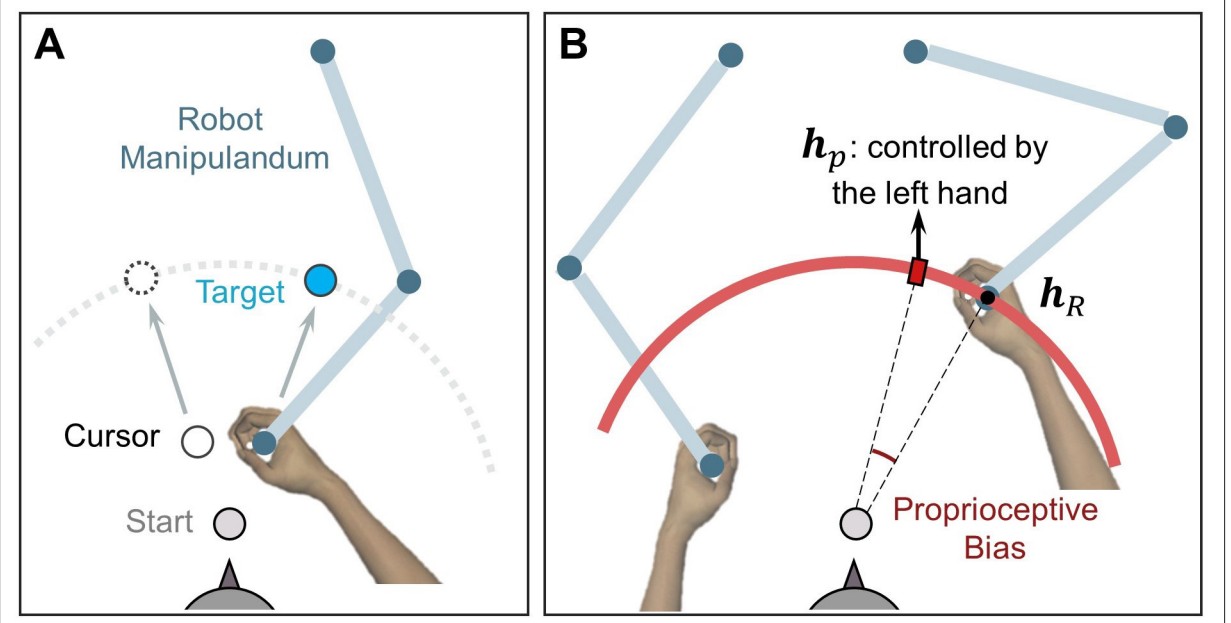

**Figure 7.** Setup for measuring proprioceptive recalibration in Experiment 3. (**A**) Reaching movement with error-clamped cursor, performed by the right hand holding a robot handle. (**B**) Passive movement in the proprioception test. The right hand was passively moved to the unseen target (h$_r$), depicted here as a small black dot. A red hollow circle with an expanding radius appears on the screen during passive movement, signaling the increasing distance between the hand and the start position. Subsequently, participants used their left hand to report the right-hand location (h$_p$) by aligning a red rectangle on the red circle, which is displayed at the target distance.

their heads. Their eye movement was recorded by an eye tracker (Tobii pro nano, Tobii, Danderyd Municipality, Sweden) affixed at the lower edge of the screen. The sampling rate was 160–200 Hz for the tablet and 60 Hz for the eye tracker.

Experiment 3 was conducted using the KINARM planar robotic manipulandum with a virtual-reality system (BKIN Technologies Ltd., Kingston, Canada). Participants, seated in a chair, held the robot handles with their left and right hands (**Figure 7**). The movement task was performed with the right handle and the left handle was used to indicate the perceived direction of the right hand in the proprioception test. A semi-silvered mirror was placed below the eye level to block the vision of the hands and the robotic manipulandum; it also served as a display monitor.

## Experiment 1: Measuring visual uncertainty in error-clamp adaptation

Eighteen among twenty participants finished the reaching with clamped error feedback and visual discrimination task in 3 consecutive days; two participants withdrew during the experiment. Participants made reaching movement by sliding the stylus from a start position at the center of the workspace to towards a target (**Figure 6A**). The start position, the target, and the cursor were represented by a gray dot, a blue cross and a white dot on the screen, respectively. All these elements had a diameter of 5 mm. The procedure of the motor and visual discrimination task is illustrated in **Figure 2A**. To initiate a trial, participants moved the cursor into the start position. Following an 800ms holding period, a target appeared 10 cm away in the twelve o'clock direction, and participants were instructed to slide through the target rapidly while maintaining a straight hand trajectory. The trial terminated when the distance between the hand and the start position exceeded 10 cm, regardless of whether the target was hit. A warning message, 'too slow', would appear on the screen if participants failed to complete the trial within 300ms after initiating the movement. Each practice day began with 60 standard reaching trials, during which veridical feedback about hand location was provided by the cursor. The target would change from blue to green if the cursor successfully passed through it. In subsequent visual clamp trials, the cursor moved along a predetermined direction set by the perturbation angle, while its position was updated in real-time based on the hand's location. The cursor's

distance from the start position was equal to the distance between the hand and the start position until the end of the trial.

Following each trial, the cursor remained frozen at its final position for an additional 800ms before disappearing. The visual discrimination task commenced 1000ms thereafter. A yellow reference point, located 10 cm from the start position, was displayed for 150ms near the cursor's final position (*Figure 2A* and *Figure 6A*). Subsequently, all visual stimuli, except for the blue cross at the start position, were removed from the screen. Participants were then required to judge whether the reference point was situated in a clockwise (CW) or counterclockwise (CCW) direction relative to the cursor's final position and to report their judgment by pressing a key on the keyboard. Participants were informed that they no longer controlled the direction of cursor movement during the task. They were instructed to fixate their gaze on either the start position or the blue cross during the motor task, while actively ignoring the white cursor. During the discrimination task, they were required to maintain their gaze on the blue cross. Eye movements were monitored in real-time using an eye tracker. Participants received a warning if their gaze was detected outside a 75-pixel-wide band-shaped region centered on the line of gaze four consecutive times during the experiment (*Figure 2—figure supplement 1*).

In each trial, the angular deviation between the error-clamped cursor and the reference point was determined using a PEST procedure (*Lieberman and Pentland, 1982*). *Figure 6C–D* illustrates the evolution of the deviation angle and step size for an exemplary participant experiencing a –16° perturbation. In each round, the deviation commenced at 30° (indicated by yellow points in *Figure 6C–D*) and was altered by one step size following each trial. The initial step size was set at 10° and was halved whenever the direction judgment changed (i.e. from 'CW' to 'CCW' or vice versa). For a specific perturbation angle, the initial deviation always started from the CW direction for the first round and flipped the direction at the beginning of the next round. A round terminated either when the step size fell below a predefined criterion (indicated by the red line in *Figure 6D*) or when the trial count exceeded 30. Six perturbation angles were randomly interleaved (*Figure 6B*), and the experiment concluded when four complete rounds of the PEST procedure had been completed for each perturbation angle. Consequently, the total number of trials varied among participants and across practice days (300–500 trials per day on average, with a maximum of 720 trials). Additionally, for some perturbation angles, more than four complete rounds could be conducted in a single day.

## Experiment 2: Motor adaptation with different perturbation size

Eighty-four participants were randomly allocated into seven groups, each comprising 12 individuals. Each group performed a motor adaptation task featuring clamped visual feedback at different perturbation angles: 2°, 4°, 8°, 16°, 32°, 64°, and 95°. As in Experiment 1, participants were instructed to slide rapidly and directly through the target, which was represented by a blue dot rather than a cross. In each trial, the target appeared at one of four possible locations (45°, 135°, 225° or 315° counter-clockwise from the positive x-axis). The sequence of target locations was randomized yet constrained so that all four positions appeared in cycles of four trials. Each group commenced with a baseline session that included 15 cycles of reaching trials with veridical feedback, followed by 15 cycles without visual feedback. Subsequently, during the perturbation session, participants completed 80 cycles of training trials featuring the error-clamped cursor with one perturbation angle (i.e. clamp size), depending on their group assignment. To assess the aftereffect, a session comprising 10 cycles of movement without visual feedback was administered. In summary, each participant practiced a total of 480 trials (120 cycles x 4 target directions) in Experiment 2.

## Experiment 3: Proprioception test with different perturbation sizes

Eleven participants were recruited for testing their proprioceptive recalibration. This experiment incorporated two types of trials: reaching trials and proprioception test trials. During the reaching trials, participants were instructed to aim for a target, which could appear at one of three possible locations (25°, 45°, or 65° counter-clockwise from the positive x-axis, as represented by light blue dots in *Figure 4C*, right panel). The task was similar to those in Experiments 1 and 2, with the key difference being that participants performed the task using KINARM robots (as depicted in *Figure 7A*). The dimensions and relative distances of the visual stimuli remained consistent with those used in Experiments 1 and 2. As in previous experiments, three kinds of visual feedback were provided during

different sessions: no visual feedback, veridical feedback, and feedback featuring an error-clamped cursor.

In the proprioception test, participants were instructed to hold the robot's right handle and wait to be passively moved by the robot to one of six proprioception targets (small red dots in *Figure 4C*, right panel). These targets were spaced at 10° intervals, ranging from 20° to 70° counterclockwise from the positive x-axis, and flanked the three reaching targets. The passive movement lasted for 1000ms and followed a straight-line path at a speed consistent with a minimum jerk velocity profile. During this movement, a ring with a 10 cm radius, centered at the start position, was displayed on the screen (depicted as a red arc in *Figure 7B*). The cursor was also replaced by a ring, its radius expanding as the hand moved toward the proprioception target.

After the right hand reached the proprioception target, participants were instructed to maintain their right hand's position. Using the left handle, they were then asked to indicate the perceived location of their right hand. The position of the left handle was mapped to the rotation of a 'dial', which was constrained to the target arc.

The position of $h_p$ was displayed on the target arc as a small red rectangle (a visual 'dial', as shown in *Figure 7B*). Participants were instructed to indicate the location of their right hand by moving the red rectangle to the position they perceived as accurate. The final position of $h_p$ was recorded when its angular velocity remained below 1 degree/second for a duration exceeding 1000ms. The proprioceptive bias was then calculated as the angular deviation between the actual hand position ($h_R$) and the perceived hand position ($h_p$).

Reaching trials and proprioception test trials were organized into blocks (*Figure 4D*). Each reaching block consisted of six trials, targeting three different locations with two repetitions each. Each reaching block was followed by a proprioception test block consisting of three trials. In these test trials, the robot moved the participant's right hand toward a target position near one of the three reaching targets. These test targets were randomly chosen from six possible locations (*Figure 4C*, right panel). The entire experiment comprised 40 reaching blocks and 40 subsequent proprioception test blocks. The first four reaching blocks provided veridical cursor feedback, the next four offered no cursor feedback, and the remaining 32 featured one of eight possible perturbation sizes ( ± 10°, ± 20°, ± 40°, and ± 80°). The size of the perturbation was randomized between blocks.

## Experiment 4: Upregulating visual uncertainty affects implicit adaptation

Nineteen participants from Experiment 1 completed Experiment 4. The reaching task employed the same setup as in Experiment 1. However, instead of performing perceptual judgments of cursor motion direction, participants engaged in movements with one of three types of cursor feedback: veridical feedback, no feedback, and feedback with clamped perturbation. To assess the influence of visual uncertainty on implicit learning, we modified the cursor to appear blurred in half of the clamped trials. The blurring mask had a diameter of 6.8 mm, and the color intensity decreased from the cursor's center following a two-dimensional Gaussian distribution with σx = σx = 1.4 mm. As depicted in *Figure 5D*, participants underwent the same procedures across three consecutive days. Each day consisted of 60 baseline trials, followed by 15 training blocks designed to assess single-trial learning. Within each training block, 12 trials featured an error-clamped cursor, each flanked by a trial without feedback. The difference between two adjacent no-feedback trials served as a measure of single-trial learning at specific perturbation sizes. Each of the 12 perturbation trials was randomly assigned one of 12 possible perturbations, comprising two cursor presentations (blurred or clear) and six clamp sizes ( ± 4°, ± 16°, ± 64°). Each participant practices 180 trials (15 blocks x 12 trials per block) with perturbation per day and 540 trials in total.

## Data analysis
### Processing of kinematic data
In Experiments 1, 2, and 4, hand kinematic data were collected online at a sampling rate ranging between 160 and 200 Hz and subsequently resampled offline to 125 Hz. The movement direction of the hand was determined by the vector connecting the start position to the hand position at the point where it crossed 50% of the target distance, that is 5 cm from the start position.

In Experiment 3, hand positions and velocities were directly acquired from the KINARM robot at a fixed sampling rate of 1 kHz. The raw kinematic data were smoothed using a fifth-order Savitzky-Golay filter with a window length of 50ms. Owing to the high temporal resolution and reliable velocity profiles provided by the KINARM system, the heading direction in Experiment 3 was calculated as the vector connecting the start position to the hand position at the point of peak velocity. For each participant, the baseline hand direction and proprioceptive bias was subtracted from the data.

## Psychometric curve

For the visual discrimination task, data of all three days were pooled together, the probability of responding that 'the reference point was in the counter-clockwise direction of the cursor' was calculated as $p$ for all angle differences (*Figure 2—figure supplement 2*). At each perturbation size, a logistic function was used to fit the probability distribution for individual participants:

$$p = \frac{1}{1 + e^{-k(x - x_0)}} \tag{5}$$

where k is the slope, and $x_0$ is the origin of the logistic function. The visual uncertainty was defined as the angle differences between 25% and 75% of the logistic function:

$$\sigma_v = \frac{loglog\ (p_2 / (1 - p_2)) - loglog\ (p_1 / (1 - p_1))}{k} \tag{6}$$

where $p_1$=25% and $p_2$=75%.

## Statistical analysis

In Experiment 1, since the visual uncertainty $\sigma_v$ follows a non-negative skewed distribution among participants, it violated the assumption of the ANOVA test. We thus applied Friedman's nonparametric test to determine whether $\sigma_v$ changes with the perturbation angle θ. Specifically, $\sigma_v$ for both positive and negative θ were subjected to Friedman's test separately, with θ serving as the factor. Spearman correlation analyses were conducted between $\sigma_v$ in different perturbation sizes. Given the symmetry between positive and negative θ, we pool the data to quantify the linear dependency of $\sigma_v$ on the absolute θ (*Equation 4*). Because $\sigma_v$ is expected to be always positive and there are potential positive outliers, we assume that it is generated from a gamma distribution rather than a normal distribution, which is always positive and favors right-skewed distribution that can well capture the positive outliers. Thus, the data was fitted by a generalized linear regression model with the absolute value of θ as independent variable and $\sigma_v$ as a dependent variable.

In Experiment 2, the adaptation extent was defined as the mean hand angles in the last 10 cycles in the perturbation phase (cycle 101–110). A one-way ANOVA with perturbation size serving as the factor to examine its influence on the adaptation extent. Pairwise post-hoc comparisons were conducted using Tukey-Kramer correction.

In Experiment 3, proprioceptive biases were quantified as the angular difference between the perceived and actual hand directions. A one-way repeated-measures ANOVA was conducted on the data of the first trial, using perturbation size as the within-subject factor. Greenhouse-Geisser corrections were applied when the assumption of sphericity was violated (*Kirk, 1968*). Multiple pairwise comparisons were conducted among different perturbation sizes for the first proprioception test. To determine if the proprioceptive biases were significantly different from zero, one-tailed (left) *t*-tests were conducted separately for the first and third proprioception test trials at each perturbation size.

In Experiment 4, the single-trial learning data was subjected to a 2 (visual uncertainty) x 3 (perturbation size) repeated-measures ANOVA. Greenhouse-Geisser corrections were applied as above, and the simple main effect of visual uncertainty was tested for each of the three perturbation sizes.

## Model fitting and simulations

### Perceptual Error Adaptation (PEA) model

Model fitting for adaptation extent as a function of perturbation size

To fit the adaptation extent data from three different experiments in previous studies in *Kim et al., 2018*; *Morehead et al., 2017*, *Equations 3 and 4* were modified for simplification. To avoid overfitting

of the small dataset, we reduced the number of model parameters by assuming that $\hat{x}_{Hand}$ asymptote to the target direction in the final adaptation trials that are used for computing adaptation extent, thus the retention rate $A \equiv 1$. Insert *Equation 4* to *Equation 3*, the asymptote hand angle with different perturbation size is:

$$x_p^{asym} = -\left( \frac{\sigma_p/a}{1 + (b/a)\,\theta} \right)^2 \theta \tag{7}$$

Two ratio parameters $R_{1,ext} = \sigma_p/a$ and $R_{2,ext} = b/a$ were used in data fitting. Three datasets were fitted separately.

## Model fitting for trial-by-trial adaptation and proprioception changes

The trial-by-trial changes of adaptation (*Figure 3A*) and of proprioceptive changes (*Figure 4A*) were fitted with *Equation 1*, *Equation 2*, and *Equation 4* based on the mean performance of all participants. The PEA model only had four free parameters, $\Theta = [\sigma_u, \sigma_p, A, B]$. The slope $a$ and intercept $b$ in *Equation 1* were obtained by psychometric tests from Experiment 1 (see statistical analysis). The reported hand position ($x_{report}$, blue dots in *Figure 4A*) was based on the proprioceptive cue $x_p$ and the estimated hand $\hat{x}_{Hand}$ from the reaching trial. With the Bayesian cue combination assumption, the reported hand position was biased by $x_p$ with a ratio determined by the variance of $x_p$ and $\hat{x}_{Hand}$:

$$x_{report} = \hat{x}_{Hand} + \frac{\sigma_{Hand}^2}{\sigma_{Hand}^2 + \sigma_p^2}\left(x_p - \hat{x}_{Hand}\right) \tag{8}$$

where $\sigma_{Hand}^2$ and $\sigma_p^2$ are the variance of $\hat{x}_{Hand}$ and $x_p$ respectively (see Appendix 1 for further details of this fitting).

To verify if the slope $b$ and intercept $a$ obtained from Experiment 1 are consistent across experiments, they were also estimated by fitting data from Experiment 2 (*Figure 3*). In this case, the model fitting was performed with 6 free parameters, $\Theta = [\sigma_u, \sigma_p, a, b, A, B]$. The fitted values of $a$ and $b$ are fallen into the 95% CI of estimated parameters in Experiment 1 (purple line in *Figure 2C*, see details in *Supplementary file 1a*).

The dependence of proprioceptive recalibration on perturbation size (*Figure 4B*) were simulated by the PEA model with the parameter values estimated from Experiment 2. We assumed that the proprioceptive bias results from the influence of a biased hand estimate ($\hat{x}_{Hand}$) during adaptation and the influence is quantified as a percentage of its deviation from the true hand location:

$$x_{bias} = -\left(0 - \hat{x}_{Hand}\right) R_p \tag{9}$$

where the actual hand location is 0, $R_p$ is the percentage of influence, and $x_{Hand}$ is determined by *Equation 1*. In simulation, $R_p$ varied from 0.05 to 0.8 to estimate the overall dependence of proprioceptive recalibration on perturbation size.

## Model fitting and simulation for single-trial learning

In the single-trial learning paradigm (*Figure 3—figure supplement 3*), the average movement direction across trials aligns with the target direction since the visual perturbations are evenly distributed in both directions. Thus, the sensory cues $x_u$ and $x_p$ have the same mean value. For modeling single-trial learning, instead of having two separate cues, we assume a combined cue of $x_u$ and $x_p$ to follow $x_{int} \sim N\left(T, \sigma_{int}^2\right)$, where $T$ is the target direction, $\sigma_{int}^2 = \frac{\sigma_u^2 \sigma_p^2}{\sigma_u^2 + \sigma_p^2}$ represents the variance of integrated sensory signal of $x_u$ and $x_p$. Single-trial learning was quantified as the difference between the two null trials before and after the perturbation trial. As the perturbation size in the triplet of trials varied randomly, we assume that the effects of different perturbations are independent. Thus, single-trial learning was modeled as learning from the current perturbation without history effect. It follows the equations modified from *Equations 1 and 2*:

$$x_{STL} = B\left(T - \hat{x}_{Hand}\right) \tag{10}$$

$$\hat{x}_{Hand} = W_{int}T + W_v x_v, \text{ with } W_{int} = \frac{1/\sigma_{int}^2}{\sum_j 1/\sigma_j^2}, \quad i,j = int, v \tag{11}$$

where $x_v$ is the visual perturbation, $W_{int}$ and $W_v$ are the weights of the cues, $\sigma_v$ is the standard deviation of the visual cue specified by *Equation 4*. Parameter set $\Theta = [\ \sigma_{int}\ ,\ a,\ b,\ B]$ was fitted to the average data from all participants. Model simulations (*Figure 5A*) were performed with the same single-trial learning equations. For the clear cursor condition, we used the same parameter values estimated from Experiment 2 (see details in *Supplementary file 1a*). For the blurred cursor condition, the standard deviation of visual cue was changed to:

$$\sigma_{v,blur} = R_v \sigma_v \tag{12}$$

for the simulation of the increase in visual uncertainty, the ratio $R_v$ varied from 1.1 to 3.

## PReMo model

We used the PReMo model to fit the average adaptation extent obtained from Experiment 2 (*Figure 3C*, *Figure 3—figure supplement 1B*). Following the study by *Tsay et al., 2022c*, the hand position at trial n+1 is:

$$x_{p,n+1} = A x_{p,n} + B \left(T - x_{per,n}\right) \tag{13}$$

where

$$x_{per,n} = \beta_p + \frac{\sigma_u^2}{\sigma_u^2 + \sigma_p^2} x_{p,n} \tag{14}$$

$$\beta_p = -min\left(\left|\beta_p^{sat}\right|, \left|\eta_p\left(\frac{\sigma_u^2}{\sigma_u^2 + \sigma_v^2} x_{v,n} - \frac{\sigma_u^2}{\sigma_u^2 + \sigma_p^2} x_{p,n}\right)\right|\right) \tag{15}$$

In data fitting, we used two parameters to represent the ratio between sensory cues: $R_1 = \frac{\sigma_u^2}{\sigma_u^2 + \sigma_v^2}$ and $R_2 = \frac{\sigma_u^2}{\sigma_p^2 + \sigma_p^2}$ . The data were fitted with the parameter set $\Theta = [\ R_1\ , R_2, \beta_p^{sat}, \eta_p,\ A,\ B]$, where $\beta_p^{sat}$ is the saturation angle, $\eta_p$ is a scaling factor, $A$ is the retention rate and $B$ is the learning rate. For simulating the proprioceptive localization of the hand (*Figure 4C*), the parameter values estimated from Experiment 2 were used. The bias of hand estimation in the proprioception trials is determined as: $x_{bias} = -\left(0 - x_{per}\right) R_p$ , where ratio $R_p$ varies from 0.05 to 0.8. Thus, similar to the PEA model simulation, the proprioceptive bias is a fraction of the bias in the hand estimation from the adaptation trials. Single-trial learning (*Figure 5B*) was simulated with:

$$x_{STL} = B \left(T - x_{per}\right) \tag{16}$$

where $x_{per}$ is determined by *Equation 12* and *Equation 13*. For the clear condition, we used the parameter values estimated from Experiment 2 with PReMo. For the blurred cursor condition, the standard deviation of visual signal $\sigma_{v,blur}$ increases with a ratio $R_v$ , as in *Equation 12*.

## Causal inference model

The causal inference model by *Wei and Körding, 2009* was used to fit the data of Experiment 2 (*Figure 3D*, *Figure 3—figure supplement 1C*). The hand position at trial n+1 is updated by learning from visual error at trial n:

$$x_{p,n+1} = A x_{p,n} + B \left(T - p x_{v,n}\right) \tag{17}$$

where $A$ and $B$ are the retention and learning rates, respectively; $T$ is the target direction. Specifically for this model, the learning from error is modulated by the probability ($p$) of causal attribution of visual error to the action or proprioception:

$$p = S \frac{N\left(x_{v,n}, 0, \sigma^2\right)}{N\left(x_{v,n}, 0, \sigma^2\right) + C} \tag{18}$$

where $x_{v,n}$ is the visual cue at trial n. $S$ and $C$ are the scaling factors, and $\sigma$ is the standard deviation of the integrated cue combining visual and proprioceptive cues, following

$$\sigma^2 = \frac{\sigma_v^2 \sigma_p^2}{\sigma_v^2 + \sigma_p^2} \tag{19}$$

Thus, the data were fitted with five parameters $\Theta = [\sigma, S, C, A, B]$. For simulating single-trial learning with cursor blurring (**Figure 5C**), the ratio between $\sigma_v$ and $\sigma_p$ is fixed as 1/2. The single-trial learning was determined as:

$$x_{STL} = B\left(T - px_v\right) \tag{20}$$

where $p$ is determined by **Equation 18**. Put **Equation 12** and **Equation 19** into $\sigma_{blur}^2 = \frac{\sigma_{v,blur}^2 \sigma_p^2}{\sigma_{v,blur}^2 + \sigma_p^2}$, we can calculate the standard deviation of the integrated sensory signal for the blurred cursor: $\sigma_{blur} = \sigma\sqrt{\frac{5R^2}{R^2+4}}$. Simulation was performed with $R$ ranging from 1.1 to 3.

## Data fitting

All data were fitted using MATLAB 2022b (MathWorks, Natick, MA, US) built-in function *fmincon* with 100 randomly sampled initial values of parameter sets. See **Supplementary file 1a** and **Supplementary file 1b** for the fitted parameter values and comparisons between different models.

## Acknowledgements

This work was supported by the STI2030-Major Project 2021ZD0202600 (2021ZD0202601) and the National Natural Science Foundation of China (62061136001, 32071047, 31871102) awarded to KW, and the National Natural Science Foundation of China (32300868) awarded to ZZ.

## Additional information

### Competing interests

Kunlin Wei: Reviewing editor, *eLife*. The other authors declare that no competing interests exist.

### Funding

| Funder | Grant reference number | Author |
|---|---|---|
| National Natural Science Foundation of China | 62061136001 | Kunlin Wei |
| National Natural Science Foundation of China | 32071047 | Kunlin Wei |
| National Natural Science Foundation of China | 31871102 | Kunlin Wei |
| National Natural Science Foundation of China | 32300868 | Zhaoran Zhang |
| STI2030-Major Project 2021ZD0202600 | 2021ZD0202601 | Kunlin Wei |

The funders had no role in study design, data collection and interpretation, or the decision to submit the work for publication.

### Author contributions

Zhaoran Zhang, Data curation, Formal analysis, Funding acquisition, Visualization, Methodology, Writing - original draft, Writing - review and editing; Huijun Wang, Data curation, Formal analysis, Methodology; Tianyang Zhang, Zixuan Nie, Data curation; Kunlin Wei, Conceptualization, Supervision, Funding acquisition, Methodology, Writing - original draft, Writing - review and editing

### Author ORCIDs

Zhaoran Zhang ⬤ http://orcid.org/0000-0002-4192-4088
Kunlin Wei ⬤ http://orcid.org/0000-0001-5098-3808

## Ethics

Participants were provided written informed consent, which was approved by the Institutional Review Board of the School of Psychological and Cognitive Sciences, Peking University.

Reviewer #1 (Public Review): https://doi.org/10.7554/eLife.94608.3.sa1
Reviewer #2 (Public Review): https://doi.org/10.7554/eLife.94608.3.sa2
Reviewer #3 (Public Review): https://doi.org/10.7554/eLife.94608.3.sa3
Author response https://doi.org/10.7554/eLife.94608.3.sa4

# Additional files

## Supplementary files

• Supplementary file 1. Model fitting results and model comparisons. (**a**) Model fitting and simulation parameters with the PEA model. (**b**) Model comparisons.
• MDAR checklist

## Data availability

Data and codes presented in this work are available at figshare and GitHub (copy archived at *Zhang, 2024*).

The following dataset was generated:

| Author(s) | Year | Dataset title | Dataset URL | Database and Identifier |
|-----------|------|---------------|-------------|-------------------------|
| Zhang Z | 2024 | Data set of the manuscript "Perceptual error based on Bayesian cue combination drives implicit motor adaptation" | https://doi.org/10.6084/m9.figshare.24503926.v2 | figshare, 10.6084/m9.figshare.24503926.v2 |

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

# Appendix 1

## Data fitting of *Figure 4A*

We use $u$, $v$, $p$ and $h$ to denote the reciprocal of the variances of motor command cue, visual cue, proprioceptive cue and combined cue (hand estimation $\hat{x}_{Hand}$) respectively:

$$u = 1/\sigma_u^2, v = 1/\sigma_v^2, p = 1/\sigma_p^2,$$

$$h = 1/\sigma_{hand}^2 = 1/\sigma_u^2 + 1/\sigma_v^2 + 1/\sigma_p^2 = u + v + p$$

According to *Equation 1* in the main text, the estimated hand can be denoted by:

$$\hat{x}_{Hand,n} = \frac{v}{h}x_v + \frac{p}{h}x_{p,n} + \frac{u}{h}x_{u,n}, \tag{A1}$$

where $x_{u,n} = 0$ indicates the target position (T). We further define the weight parameters $W_v$ and $W_p$ as:

$$W_v = \frac{v}{h}$$

$$W_p = \frac{p}{h}$$

and *Equation A1* can be wrote as:

$$\hat{x}_{Hand,n} = W_v x_v + W_p x_{p,n}, \tag{A2}$$

insert *Equation A2* into *Equation 2* from the main text, the hand position at trial n is:

$$x_{p,n+1} = A x_{p,n} + B\left(0 - W_v x_v + W_p x_{p,n}\right) \tag{A3}$$

Next, with the same Bayesian cue combination assumption, the reported hand position was combined with $x_p$ and $\hat{x}_{Hand}$:

$$x_{Report,n} = W_{\widehat{Hand}}\hat{x}_{Hand,n} + W_{p*} x_{p,n}, \tag{A4}$$

where $W_{\widehat{Hand}}$ and $W_{p*}$ are the weight parameters:

$$W_{\widehat{Hand}} = \frac{h}{h + p} = \frac{1}{1 + W_p}$$

$$W_{p*} = \frac{p}{h + p} = \frac{W_p}{1 + W_p}$$

put the weight parameters and *Equation A2* into *Equation A4*, we can easily get:

$$x_{Report,n} = \frac{W_v}{1 + W_p}x_v + \frac{2W_p}{1 + W_p}x_{p,n} \tag{A5}$$

The adaptation epoch of the actual and reported hand position in *Figure 4A* was fitted by *Equation A3* and *Equation A5* respectively with the parameter set $\Theta = [W_v, W_p, A, B]$.

We simulated the wash out epoch based on the parameters estimated from the adaptation epoch with the method below (dashed line in *Figure 4A*).

Since the visual cue was no longer available, the weight of cues changed to:

$$W_{v[wo]} = 0,$$

$$W_{p[wo]} = \frac{p}{u + p} = \frac{p}{h - v} = \frac{W_p}{1 - W_v}.$$

Since $x_{u,n} = 0$, the estimated hand in wash out epoch is:

$$\hat{x}_{Hand,n} = W_{v[wo]} x_v + W_{p[wo]} x_{p,n} \tag{A6}$$

put $W_{v[wo]} W_{p[wo]}$ and *Equation A6* into *Equation 2* in the main text, the hand position at trial n is:

$$x_{p,n+1} = A x_{p,n} + B \left( 0 - \frac{W_p}{1 - W_v} x_{p,n} \right) \tag{A7}$$

Similar to *Equation A4*, the reported hand position is:

$$x_{Report,n} = W_{\widehat{Hand}[wo]} \hat{x}_{Hand,n} + W_{p^*[wo]} x_{p,n} \tag{A8}$$

where $W_{\widehat{Hand}[wo]}$ and $W_{p^*[wo]}$ denote the corresponding weight of each cue:

$$W_{\widehat{Hand}[wo]} = \frac{h_{[wo]}}{h_{[wo]} + p} = \frac{1 - W_v}{1 - W_v + W_p}$$

$$W_{p^*[wo]} = \frac{p}{h_{[wo]} + p} = \frac{W_p}{1 - W_v + W_p}$$

where $h_{[wo]}$ denotes the reciprocal of the variances of estimated hand during washout epoch:

$$h_{[wo]} = u + p = h - v$$

Put the weight parameters and A6 into A8, the reported hand position can be easily derived:

$$x_{Report,n} = \left( \frac{W_p (1 - W_v)}{(1 - W_v + W_p)(1 - W_v)} + \frac{W_p}{1 - W_v + W_p} \right) x_{p,n} \tag{A9}$$

*Equation A7 and A9* were used to simulate the hand position and report position during washout epoch with $\Theta = [W_v, W_p, A, B]$ estimated from the adaptation epoch.

# Appendix 2

## Fitting block designed adaptation data (Experiment 2) without the assumption of linearity in visual uncertainty

For each perturbation size $i$, the weight of visual and proprioceptive cues are:

$$W_v^{[i]} = \frac{1/\sigma_{v[i]}^2}{1/\sigma_{u[i]}^2 + 1/\sigma_{v[i]}^2 + 1/\sigma_{p[i]}^2}, \tag{A10}$$

$$W_p^{[i]} = \frac{1/\sigma_{p[i]}^2}{1/\sigma_{u[i]}^2 + 1/\sigma_{v[i]}^2 + 1/\sigma_{p[i]}^2}, i = 1 \cdots 7 \tag{A11}$$

where $\sigma_{u[i]}$, $\sigma_{v[i]}$ and $\sigma_{p[i]}$ are the uncertainties of sensory cues under different perturbation sizes (2–95 degree). The estimated hand can be calculated by putting *Equation A10* and *Equation A11* into *Equation 1* from the main text:

$$\hat{x}_{Hand,n} = W_v^{[i]} x_v + W_p^{[i]} x_{p,n} + \frac{1/\sigma_{u[i]}^2}{1/\sigma_{u[i]}^2 + 1/\sigma_{v[i]}^2 + 1/\sigma_{p[i]}^2} x_{u,n}$$

where $x_{u,n} = T = 0$ denotes the target location, thus:

$$\hat{x}_{Hand,n} = W_v^{[i]} x_v + W_p^{[i]} x_{p,n} \tag{A12}$$

Insert *Equation A12* into *Equation 2* from the main text, the trial-by-trail hand position is:

$$x_{p,n+1} = -BW_v^{[i]} x_v + \left( A - BW_p^{[i]} \right) x_{p,n} \tag{A13}$$

The parameter set $\Theta = [W_v^{[1]} \ldots W_v^{[7]}, W_p^{[1]} \ldots W_p^{[7]}, A, B]$ were fitted to the data in Experiment 2. Data fitting was done with a Bayesian optimization algorithm (*Acerbi and Ma, 2017*) with 100 runs of different initial values between a set of predetermined preferred boundaries.

The ratio between uncertainties of different sensory cues are:

$$\sigma_{v[i]}/\sigma_{u[i]} = \sqrt{\left( 1 - W_v^{[i]} - W_p^{[i]} \right) / W_v^{[i]}} \tag{A14}$$

$$\sigma_{p[i]}/\sigma_{u[i]} = \sqrt{\left( 1 - W_v^{[i]} - W_p^{[i]} \right) / W_p^{[i]}} \tag{A15}$$

The fitting results are presented in *Figure 3—figure supplement 4A*. Assuming $\sigma_u$ remains constant across different perturbation sizes, we convert $\sigma_p$ and $\sigma_v$ into ratios, *Figure 3—figure supplement 4B* illustrates the ratio between estimated uncertainties at different perturbation sizes.

To determine the values shown in *Figure 3—figure supplement 4C*, we proceeded as follows: First, we found no significant correlation between perturbation sizes and the estimated values of $\sigma_p$ ($R=-0.476$, $P=0.281$). The mean of $\sigma_p$ is thus set to the mean value derived from the proprioception uncertainty experiment (*Figure 4—figure supplement 1*), with $\sigma_p = 9.737°$ for all perturbation sizes, and $\sigma_u = 3.681°$ was also calculated as a ratio of $\sigma_p$. Those results are indicated by the two horizontal dashed lines in *Figure 3—figure supplement 4C*. Subsequently, we estimated $\sigma_v$ for different perturbation sizes, represented by dark brown dots in *Figure 3—figure supplement 4C*. To compare the visual uncertainty estimated from Experiment 2 with the results from Experiment 1, we performed a linear fit of the estimated $\sigma_v$ values and perturbation sizes using first-order polynomials. The fitting yielded an intercept a=3.617 and a slope b=0.261 ($R^2=0.982$, $P<0.001$). The slope and intercept are in close agreement with those from Experiment 1 (the gray line in *Figure 3—figure supplement 4C*), and the fitting results also confirmed a strong linear relationship between $\sigma_v$ and perturbation sizes.

