## [Editor Report · eLife assessment]

This study presents an **important** finding on the influence of visual uncertainty and Bayesian cue combination on implicit motor adaptation in young healthy participants, hereby linking perception and action during implicit adaptation. The evidence supporting the claims of the authors is **convincing**. The normative approach of the proposed PEA model, which combines ideas from separate lines of research, including vision research and motor learning, opens avenues for future developments. This work will be of interest to researchers in sensory cue integration and motor learning.

---

## [Referee Report · Reviewer #1 (Public Review)]

I appreciate the normative approach of the PEA model and am eager to examine this model in the future. However, two minor issues remain:

(1) Clarification on the PReMo Model:

The authors state, "The PReMo model proposes that this drift comprises two phases: initial proprioceptive recalibration and subsequent visual recalibration." This description could misinterpret the intent of PReMo. According to PReMo, the time course of the reported hand position is merely a read-out of the *perceived hand position* (x_hat in your paper). Early in adaptation, the perceived hand position is biased by the visual cursor (x_hat in the direction of the cursor); towards the end, due to implicit adaptation, x_hat reduces to zero. This is the same as PEA. I recommend that the authors clarify PReMo's intent to avoid confusion.

Note, however, the observed overshoot of 1 degree in the reported hand position. In the PReMo paper, we hypothesized that this effect is due to the recalibration of the perceived visual target location (inspired by studies showing that vision is also recalibrated by proprioception, but in the opposite direction). If the goal of implicit adaptation is to align the perceived hand position (x_hat) with the perceived target position (t_hat), then there would be an overshoot of x_hat over the actual target position.

PEA posits a different account for the overshoot. It currently suggests that the reported hand position combines x_hat (which takes x_p as input) with x_p itself. What is reasoning underlying the *double occurrence* of x_p?

There seem to be three alternatives that seem more plausible (and could lead to the same overshooting): (1) increasing x_p's contribution (assuming visual uncertainty increases when the visual cursor is absent during the hand report phase), (2) decreasing sigma_p (assuming that participants pay more attention to the hand during the report phase), (3) it could be that the perceived target position undergoes recalibration in the opposite direction to proprioceptive recalibration. All these options, at least to me, seem equally plausible and testable in the future.

(2) Effect of Visual Uncertainty on Error Size:

I appreciate the authors' response about methodological differences between the cursor cloud used in previous studies and the Gaussian blob used in the current study. However, it is still not clear to me how the authors reconcile previous studies showing that visual uncertainty reduced implicit adaptation for small but not large errors (Tsay et al, 2021; Makino, et al 2023) with the current findings, where visual uncertainty reduced implicit adaptation for large but not small errors.

Could the authors connect the dots here: I could see that the cursor cloud increases potential overlap with the visual target when the visual error is small, resulting in intrinsic reward-like mechanisms (Kim et al, 2019), which could potentially explain attenuated implicit adaptation for small visual errors. However, why would implicit adaptation in response to large visual errors remain unaffected by the cursor cloud? Note that we did verify that sigma_v is increased in (Tsay et al. 2021), so it is unlikely due to the cloud simply failing as a manipulation of visual uncertainty.

In addition, we also reasoned that testing individuals with low vision could offer a different test of visual uncertainty (Tsay et al, 2023). The advantage here is that both control and patients with low vision are provided with the same visual input-a single cursor. Our findings suggest that uncertainty due to low vision also shows reduced implicit adaptation in response to small but not large errors, contrary to the findings in the current paper. Missing in the manuscript is a discussion related to why the authors' current findings contradict those of previous results.

---

## [Referee Report · Reviewer #2 (Public Review)]

Summary:

The authors present the Perceptual Error Adaptation (PEA) model, a computational approach offering a unified explanation for behavioral results that are inconsistent with standard state-space models. Beginning with the conventional state-space framework, the paper introduces two innovative concepts. Firstly, errors are calculated based on the perceived hand position, determined through Bayesian integration of visual, proprioceptive, and predictive cues. Secondly, the model accounts for the eccentricity of vision, proposing that the uncertainty of cursor position increases with distance from the fixation point. This elegantly simple model, with minimal free parameters, effectively explains the observed plateau in motor adaptation under the implicit motor adaptation paradigm using the error-clamp method. Furthermore, the authors experimentally manipulate visual cursor uncertainty, a method established in visuomotor studies, to provide causal evidence. Their results show that the adaptation rate correlates with perturbation sizes and visual noise, uniquely explained by the PEA model and not by previous models. Therefore, the study convincingly demonstrates that implicit motor adaptation is a process of Bayesian cue integration

Strengths:

In the past decade, numerous perplexing results in visuomotor rotation tasks have questioned their underlying mechanisms. Prior models have individually addressed aspects like aiming strategies, motor adaptation plateaus, and sensory recalibration effects. However, a unified model encapsulating these phenomena with a simple computational principle was lacking. This paper addresses this gap with a robust Bayesian integration-based model. Its strength lies in two fundamental assumptions: motor adaptation's influence by visual eccentricity, a well-established vision science concept, and sensory estimation through Bayesian integration. By merging these well-founded principles, the authors elucidate previously incongruent and diverse results with an error-based update model. The incorporation of cursor feedback noise manipulation provides causal evidence for their model. The use of eye-tracking in their experimental design, and the analysis of adaptation studies based on estimated eccentricity, are particularly elegant. This paper makes a significant contribution to visuomotor learning research.

The authors discussed in the revised version that the proposed model can capture the general implicit motor learning process in addition to the visuomotor rotation task. In the discussion, they emphasize two main principles: the automatic tracking of effector position and the combination of movement cues using Bayesian integration. These principles are suggested as key to understanding and modeling various motor adaptations and skill learning. The proposed model could potentially become a basis for creating new computational models for skill acquisition, especially where current models fall short.

Weaknesses:

The proposed model is described as elegant. In this paper, the authors test the model within a limited example condition, demonstrating its relevance to the sensorimotor adaptation mechanisms of the human brain. However, the scope of the model's applicability remains unclear. It has shown the capacity to explain prior data, thereby surpassing previous models that rely on elementary mathematics. To solidify its credibility in the field, the authors must gather more supporting evidence.

---

## [Referee Report · Reviewer #3 (Public Review)]

(2.1) Summary

In this paper, the authors model motor adaptation as a Bayesian process that combines visual uncertainty about the error feedback, uncertainty about proprioceptive sense of hand position, and uncertainty of predicted (=planned) hand movement with a learning and retention rate as used in state space models. The model is built with results from several experiments presented in the paper and is compared with the PReMo model (Tsay, Kim et al., 2022) as well as a cue combination model (Wei & Körding, 2009). The model and experiments demonstrate the role of visual uncertainty about error feedback in implicit adaptation.

In the introduction, the authors notice that implicit adaptation (as measured in error-clamp based paradigms) does not saturate at larger perturbations, but decreases again (e.g. Moorehead et al., 2017 shows no adaptation at 135{degree sign} and 175{degree sign} perturbations). They hypothesized that visual uncertainty about cursor position increases with larger perturbations since the cursor is further from the fixated target. This could decrease importance assigned to visual feedback which could explain lower asymptotes.

The authors characterize visual uncertainty for 3 rotation sizes in a first experiment, and while this experiment could be improved, it is probably sufficient for the current purposes. Then the authors present a second experiment where adaptation to 7 clamped errors are tested in different groups of participants. The models' visual uncertainty is set using a linear fit to the results from experiment 1, and the remaining 4 parameters are then fit to this second data set. The 4 parameters are (1) proprioceptive uncertainty, (2) uncertainty about the predicted hand position, (3) a learning rate and (4) a retention rate. The authors' Perceptual Error Adaptation model ("PEA") predicts asymptotic levels of implicit adaptation much better than both the PReMo model (Tsay, Kim et al., 2022), which predicts saturated asymptotes, or a causal inference model (Wei & Körding, 2007) which predicts no adaptation for larger rotations. In a third experiment, the authors test their model's predictions about proprioceptive recalibration, but unfortunately compare their data with an unsuitable other data set (Tsay et al. 2020, instead of Tsay et al. 2021). Finally, the authors conduct a fourth experiment where they put their model to the test. They measure implicit adaptation with increased visual uncertainty, by adding blur to the cursor, and the results are again better in line with their model (predicting overall lower adaptation), than with the PReMo model (predicting equal saturation but at larger perturbations) or a causal inference model (predicting equal peak adaptation, but shifted to larger rotations). In particular the model fits for experiment 2 and the results from experiment 4 show that the core idea of the model has merit: increased visual uncertainty about errors dampens implicit adaptation.

(2.2) Strengths

In this study the authors propose a Perceptual Error Adaptation model ("PEA") and the work combines various ideas from the field of cue combination, Bayesian methods and new data sets, collected in four experiments using various techniques that test very different components of the model. The central component of visual uncertainty is assessed in a first experiment. The model uses 4 other parameters to explain implicit adaptation. These parameters are: (1) a learning and (2) a retention rate, as used in popular state space models and the uncertainty (variance) of (3) predicted and (4) proprioceptive hand position. In particular, the authors observe that asymptotes for implicit learning do not saturate, as claimed before, but decrease again when rotations are very large and that this may have to do with visual uncertainty (e.g. Tsay et al., 2021, J Neurophysiol 125, 12-22). The final experiment confirms predictions of the fitted model about what happens when visual uncertainty is increased (overall decrease of adaptation). By incorporating visual uncertainty depending on retinal eccentricity, the predictions of the PEA model for very large perturbations are notably different from, and better than, the predictions of the two other models it is compared to. That is, the paper provides strong support for the idea that visual uncertainty of errors matters for implicit adaptation.

(2.3) Weaknesses

Although the authors don't say this, the "concave" function that shows that adaptation does not saturate for larger rotations has been shown before, including in papers cited in this manuscript.

The first experiment, measuring visual uncertainty for several rotation sizes in error-clamped paradigms has several shortcomings, but these might not be so large as to invalidate the model or the findings in the rest of the manuscript. There are two main issues we highlight here. First, the data is not presented in units that allow comparison with vision science literature. Second, the 1 second delay between movement endpoint and disappearance of the cursor, and the presentation of the reference marker, may have led to substantial degradation of the visual memory of the cursor endpoint. That is, the experiment could be overestimating the visual uncertainty during implicit adaptation.

The paper's third experiment relies to a large degree on reproducing patterns found in one particular paper, where the reported hand positions - as a measure of proprioceptive sense of hand position - are given and plotted relative to an ever present visual target, rather than relative to the actual hand position. That is, (1) since participants actively move to a visual target, the reported hand positions do not reflect proprioception, but mostly the remembered position of the target participants were trying to move to, and (2) if the reports are converted to a difference between the real and reported hand position (rather than the difference between the target and the report), those would be on the order of ~20{degree sign} which is roughly two times larger than any previously reported proprioceptive recalibration, and an order of magnitude larger than what the authors themselves find (1-2{degree sign}) and what their model predicts. Experiment 3 is perhaps not crucial to the paper, but it nicely provides support for the idea that proprioceptive recalibration can occur with error-clamped feedback.

Perhaps the largest caveat to the study is that it assumes that people do not look at the only error feedback available to them (and can explicitly suppress learning from it). This was probably true in the experiments used in the manuscript, but unlikely to be the case in most of the cited literature. Ignoring errors and suppressing adaptation would also be a disastrous strategy to use in the real world, such that our brains may not be very good at this. So the question remains to what degree - if any - the ideas behind the model generalize to experiments without fixation control, and more importantly, to real life situations.

---

## [Author Response]

The following is the authors’ response to the current reviews.

**eLife assessment**
This study presents an important finding on the influence of visual uncertainty and Bayesian cue combination on implicit motor adaptation in young healthy participants, hereby linking perception and action during implicit adaptation. The evidence supporting the claims of the authors is convincing. The normative approach of the proposed PEA model, which combines ideas from separate lines of research, including vision research and motor learning, opens avenues for future developments. This work will be of interest to researchers in sensory cue integration and motor learning.

Thank you for the updated assessment. We are also grateful for the insightful and constructive comments from the reviewers, which have helped us improve the manuscript again. We made necessary changes following their comments (trimmed tests, new analysis results, etc) and responded to the comments in a point-by-point fashion below. We hope to publish these responses alongside the public review. Thank you again for fostering the fruitful discussion here.

**Public Reviews:**

**Reviewer #1 (Public Review):**
I appreciate the normative approach of the PEA model and am eager to examine this model in the future. However, two minor issues remain:(1) Clarification on the PReMo Model:The authors state, "The PReMo model proposes that this drift comprises two phases: initial proprioceptive recalibration and subsequent visual recalibration." This description could misinterpret the intent of PReMo. According to PReMo, the time course of the reported hand position is merely a read-out of the *perceived hand position* (x_hat in your paper). Early in adaptation, the perceived hand position is biased by the visual cursor (x_hat in the direction of the cursor); towards the end, due to implicit adaptation, x_hat reduces to zero. This is the same as PEA. I recommend that the authors clarify PReMo's intent to avoid confusion.Note, however, the observed overshoot of 1 degree in the reported hand position. In the PReMo paper, we hypothesized that this effect is due to the recalibration of the perceived visual target location (inspired by studies showing that vision is also recalibrated by proprioception, but in the opposite direction). If the goal of implicit adaptation is to align the perceived hand position (x_hat) with the perceived target position (t_hat), then there would be an overshoot of x_hat over the actual target position.PEA posits a different account for the overshoot. It currently suggests that the reported hand position combines x_hat (which takes x_p as input) with x_p itself. What is reasoning underlying the *double occurrence* of x_p?There seem to be three alternatives that seem more plausible (and could lead to the same overshooting): (1) increasing x_p's contribution (assuming visual uncertainty increases when the visual cursor is absent during the hand report phase), (2) decreasing sigma_p (assuming that participants pay more attention to the hand during the report phase), (3) it could be that the perceived target position undergoes recalibration in the opposite direction to proprioceptive recalibration. All these options, at least to me, seem equally plausible and testable in the future.

For clarification of the PReMo model’s take on Fig4A, we now write:

“The PReMo model proposes that the initial negative drift reflects a misperceived hand location, which gradually reduces to zero, and the late positive drift reflects the influence of visual calibration of the target (Tsay, Kim, Saxena, et al., 2022). ”

However, we would like to point out that the PEA model does not predict a zero x^hand  (perceived hand location) even at the late phase of adaptation: it remains negative, though not as large as during initial adaptation (see Figure 4A, red line). Furthermore, we have not seen any plausible way to use a visually biased target to explain the overshoot of the judged hand location (see below when we address the three alternative hypotheses the reviewer raised).

We don’t think the “double” use of *xp* is a problem, simply because there are TWO tasks under investigation when the proprioceptive changes are measured along with adaptation. The first is the reaching adaptation task itself: moving under the influence of the clamped cursor. This task is accompanied by a covert estimation of hand location after the movement (x^hand ). Given the robustness of implicit adaptation, this estimation appears mandatory and automatic. The second task is the hand localization task, during which the subject is explicitly asked to judge where the hand is. Here, the perceived hand is based on the two available cues, one is the actual hand location *xp*, and the other is the influence from the just finished reaching movement (i.e., x^hand ). For Bayesian modeling from a normative perspective, sensory integration is based on the available cues to fulfill the task. For the second task of reporting the hand location, the two cues are *xp* and *xp* is used sequentially in this sense. Thus, its dual use is well justified.x^hand  (with a possible effect of the visual target, which is unbiased since it is defined as 0 in model simulation; thus, its presence does not induce any shift effect).

Our hypothesis is that the reported hand position results from a combination of x^hand  from the previous movement and the current hand position *xp*. However, specifically for the overshoot of the judged hand location in the late part of the adaptation (Fig4A), the reviewer raised three alternative explanations by assuming that the PReMo model is correct. Under the PReMo model, the estimated hand location is only determined by x^hand , and *xp* is not used in the hand location report phase. In addition, *xp* used once) and a visual recalibration of the target can explain away the gradual shift from negative to positive (overshoot).x^hand  (with

We don’t think any of them can parsimoniously explain our findings here, and we go through these three hypotheses one by one:

(1) increasing *xp*'s contribution (assuming visual uncertainty increases when the visual cursor is absent during the hand report phase)

(2) decreasing *σp* (assuming that participants pay more attention to the hand during the report phase)

The first two alternative explanations basically assume that *xp* has a larger contribution (weighting in Bayesian terms) in the hand location report phase than in the adaptation movement phase, no matter due to an increase in visual uncertainty (alternative explanation 1) or a reduction in proprioceptive uncertainty (alternative explanation 2). Thus, we assume that the reviewer suggests that a larger weight for *xp* can explain why the perceived hand location changes gradually from negative to positive. However, per the PReMo model, a larger weight for the *xp* will only affect x^hand , which is already assumed to change from negative to zero. More weight in in the hand report phase (compared to the adaptation movement phase) would not explain away the reported hand location from negative to positive. This is because no matter how much weight the *xp* has, the PReMo model assumes a saturation (βpsat ) for the influence of *xp* on x^hand . Thus x^hand  would not exceed zero in the late adaptation. Then, the PReMo model would rely on the so-called visual shift of the target to explain the overshoot. This leads us to the third alternative the reviewer raised:

(3) it could be that the perceived target position undergoes recalibration in the opposite direction to proprioceptive recalibration.

The PReMo model originally assumed that the perceived *target* location was biased in order to explain away the positive overshoot of the reported *hand* location. We assume that the reviewer suggests that the perceived target position, which is shifted to the positive direction, also “biases” the perceived hand position. We also assume that the reviewer suggests that the perceived hand location after a clamp trial (x^hand ) is zero, and somehow the shifted perceived target position “biases” the reported hand location after a clamp trial. Unfortunately, we did not see any mathematical formulation of this biasing effect in the original paper (Tsay, Kim, Haith, et al., 2022). We are not able to come up with any formulation of this hypothesized biasing effect based on Bayesian cue integration principles. Target and hand are two separate perceived items; how one relates to another needs justification from a normative perspective when discussing Bayesian models. Note this is not a problem for our PEA models, in which both cues used are about hand localization, one is x^hand  and the other is *xp*.

We believe that mathematically formulating the biasing effect (Figure 4A) is non-trivial since the reported hand location changes continuously from negative to positive. Thus, quantitative model predictions, like the ones our PEA model presents here, are needed.

To rigorously test the possible effect of visual recalibration of the target, there are two things to do: (1) use the psychometric method to measure the biased perception of the target, and (2) re-do Tsay et al. 2020 experiment without the target. For (2), compared to the case with the target, the PEA model would predict a larger overshoot, while the PReMo would predict a smaller overshoot or even zero overshoot. This can be left for future studies.

(2) Effect of Visual Uncertainty on Error Size:I appreciate the authors' response about methodological differences between the cursor cloud used in previous studies and the Gaussian blob used in the current study. However, it is still not clear to me how the authors reconcile previous studies showing that visual uncertainty reduced implicit adaptation for small but not large errors (Tsay et al, 2021; Makino, et al 2023) with the current findings, where visual uncertainty reduced implicit adaptation for large but not small errors.Could the authors connect the dots here: I could see that the cursor cloud increases potential overlap with the visual target when the visual error is small, resulting in intrinsic reward-like mechanisms (Kim et al, 2019), which could potentially explain attenuated implicit adaptation for small visual errors. However, why would implicit adaptation in response to large visual errors remain unaffected by the cursor cloud? Note that we did verify that sigma_v is increased in (Tsay et al. 2021), so it is unlikely due to the cloud simply failing as a manipulation of visual uncertainty.In addition, we also reasoned that testing individuals with low vision could offer a different test of visual uncertainty (Tsay et al, 2023). The advantage here is that both control and patients with low vision are provided with the same visual input-a single cursor. Our findings suggest that uncertainty due to low vision also shows reduced implicit adaptation in response to small but not large errors, contrary to the findings in the current paper. Missing in the manuscript is a discussion related to why the authors' current findings contradict those of previous results.

For connecting the dots for two previous studies (Tsay et al., 2021, 2023); Note Makino et al., 2023 is not in this discussion since it investigated the weights of multiple cursors, as opposed to visual uncertainty associated with a cursor cloud:

First, we want to re-emphasize that using the cursor cloud to manipulate visual uncertainty brings some confounds, making it not ideal for studying visuomotor adaptation. For example, in the error clamp paradigm, the error is defined as angular deviation. The cursor cloud consists of multiple cursors spanning over a range of angles, which affects both the sensory uncertainty (the intended outcome) and the sensory estimate of angles (the error estimate, the undesired outcome). In Bayesian terms, the cursor cloud aims to modulate the sigma of a distribution (*σv*) in our model, but it additionally affects the mean of the distribution (µ). This unnecessary confound is neatly avoided by using cursor blurring, which is still a cursor with its center (µ) unchanged from a single cursor. Furthermore, as correctly pointed out in the original paper by Tsay et al., 2020, the cursor cloud often overlaps with the visual target; this "target hit" would affect adaptation, possibly via a reward learning mechanism (Kim et al., 2019). This is a second confound that accompanies the cursor cloud. Yes, the cursor cloud was verified as associated with high visual uncertainty (Tsay et al., 2021); this verification was done with a psychophysics method with a clean background, not in the context of a hand reaching a target that is needed. Thus, despite the cursor cloud having a sizeable visual uncertainty, our criticisms for it still hold when used in error-clamp adaptation.

Second, bearing these confounds of the cursor cloud in mind, we postulate one important factor that has not been considered in any models thus far that might underlie the lack of difference between the single-cursor clamp and the cloud-cursor clamp when the clamp size is large: the cursor cloud might be harder to ignore than a single cursor. For Bayesian sensory integration, the naive model is to consider the relative reliability of cues only. Yes, the cloud is more uncertain in terms of indicating the movement direction than a single cursor. However, given its large spread, it is probably harder to ignore during error-clamp movements. Note that ignoring the clamped cursor is the task instruction, but the large scatter of the cursor cloud is more salient and thus plausible and harder to ignore. This might increase the weighting of the visual cue despite its higher visual uncertainty. This extra confound is arguably minimized by using the blurred cursor as in our Exp4 since the blurred cursor did not increase the visual angle much (Figure 5D; blurred vs single cursor: 3.4mm vs 2.5mm in radius, 3.90o vs 2.87o in spread). In contrast, the visual angle of the dot cloud is at least a magnitude larger (cursor cloud vs. single cursor: at least 25o vs. 2.15o in the spread, given a 10o standard deviation of random sampling).

Third, for the low-vision study (Tsay et al., 2023), the patients indeed show reduced implicit adaptation for a 3 o clamp (consistent with our PEA model) but an intact adaptation for 30-degree clamp (not consistent). Though this pattern appears similar to what happens for normal people whose visual uncertainty is upregulated by cursor cloud (Tsay et al., 2021), we are not completely convinced that the same underlying mechanism governs these two datasets. Low-vision patients indeed have higher visual uncertainty about color, brightness, and object location, but their visual uncertainty about visual motion is still unknown. Due to the difference in impairment among low vision people (e.g., peripheral or central affected) and the different roles of peripheral and central vision in movement planning and control (Sivak & Mackenzie, 1992), it is unclear about the overall effect of visual uncertainty in low vision people. The direction of cursor movement that matters for visuomotor rotation here is likely related to visual motion perception. Unfortunately, the original study did not measure this uncertainty in low-vision patients. We believe our Exp1 offers a valid method for this purpose for future studies. More importantly, we should not expect low-vision patients to integrate visual cues in the same way as normal people, given their long-term adaptation to their vision difficulties. Thus, we are conservative about interpreting the seemingly similar findings across the two studies (Tsay et al., 2021, 2023) as revealing the same mechanism.

A side note: these two previous studies proposed a so-called mis-localization hypothesis, i.e., the cursor cloud was mislocated for small clamp size (given its overlapping with the target) but not for large clamp size. They suggested that the lack of uncertainty effect at small clamp sizes is due to mislocalization, while the lack of uncertainty effect at large clamp sizes is because implicit adaptation is not sensitive to uncertainty at large angles. Thus, these two studies admit that cursor cloud not only upregulates uncertainty but also generates an unwanted effect of so-called “mis-localization” (overlapping with the target). Interestingly, their hypothesis about less sensitivity to visual uncertainty for large clamps is not supported by a model or theory but merely a re-wording of the experiment results.

In sum, our current study cannot offer an easy answer to "connect the dots" in the aforementioned two studies due to methodology issues and the specialty of the population. However, for resolving conflicting findings, our study suggests solutions include using a psychometric test to quantify visual uncertainty for cursor motion (Exp1), a better uncertainty-manipulation method to avoid a couple of confounds (Exp4, blurred cursor), and a falsifiable model. Future endeavors can solve the difference between studies based on the new insights from the current.

**Reviewer #2 (Public Review):**
Summary:The authors present the Perceptual Error Adaptation (PEA) model, a computational approach offering a unified explanation for behavioral results that are inconsistent with standard state-space models. Beginning with the conventional state-space framework, the paper introduces two innovative concepts. Firstly, errors are calculated based on the perceived hand position, determined through Bayesian integration of visual, proprioceptive, and predictive cues. Secondly, the model accounts for the eccentricity of vision, proposing that the uncertainty of cursor position increases with distance from the fixation point. This elegantly simple model, with minimal free parameters, effectively explains the observed plateau in motor adaptation under the implicit motor adaptation paradigm using the error-clamp method. Furthermore, the authors experimentally manipulate visual cursor uncertainty, a method established in visuomotor studies, to provide causal evidence. Their results show that the adaptation rate correlates with perturbation sizes and visual noise, uniquely explained by the PEA model and not by previous models. Therefore, the study convincingly demonstrates that implicit motor adaptation is a process of Bayesian cue integrationStrengths:In the past decade, numerous perplexing results in visuomotor rotation tasks have questioned their underlying mechanisms. Prior models have individually addressed aspects like aiming strategies, motor adaptation plateaus, and sensory recalibration effects. However, a unified model encapsulating these phenomena with a simple computational principle was lacking. This paper addresses this gap with a robust Bayesian integration-based model. Its strength lies in two fundamental assumptions: motor adaptation's influence by visual eccentricity, a well-established vision science concept, and sensory estimation through Bayesian integration. By merging these well-founded principles, the authors elucidate previously incongruent and diverse results with an error-based update model. The incorporation of cursor feedback noise manipulation provides causal evidence for their model. The use of eye-tracking in their experimental design, and the analysis of adaptation studies based on estimated eccentricity, are particularly elegant. This paper makes a significant contribution to visuomotor learning research.The authors discussed in the revised version that the proposed model can capture the general implicit motor learning process in addition to the visuomotor rotation task. In the discussion, they emphasize two main principles: the automatic tracking of effector position and the combination of movement cues using Bayesian integration. These principles are suggested as key to understanding and modeling various motor adaptations and skill learning. The proposed model could potentially become a basis for creating new computational models for skill acquisition, especially where current models fall short.Weaknesses:The proposed model is described as elegant. In this paper, the authors test the model within a limited example condition, demonstrating its relevance to the sensorimotor adaptation mechanisms of the human brain. However, the scope of the model's applicability remains unclear. It has shown the capacity to explain prior data, thereby surpassing previous models that rely on elementary mathematics. To solidify its credibility in the field, the authors must gather more supporting evidence.

Indeed, our model here is based on one particular experimental paradigm, i.e., the error-clamp adaptation. We used it simply because (1) this paradigm is one rare example that implicit motor learning can be isolated in a clean way, and (2) there are a few conflicting findings in the literature for us to explain away by using a unified model.

For our model’s broad impact, we believe that as long as people need to locate their effectors during motor learning, the general principle laid out here will be applicable. In other words, repetitive movements with a Bayesian cue combination of movement-related cues can underlie the implicit process of various motor learning. To showcase its broad impact, in upcoming studies, we will extend this model to other motor learning paradigms, starting from motor adaptation paradigms that involve both explicit and implicit processes.

**Reviewer #3 (Public Review):**
(2.1) SummaryIn this paper, the authors model motor adaptation as a Bayesian process that combines visual uncertainty about the error feedback, uncertainty about proprioceptive sense of hand position, and uncertainty of predicted (=planned) hand movement with a learning and retention rate as used in state space models. The model is built with results from several experiments presented in the paper and is compared with the PReMo model (Tsay, Kim et al., 2022) as well as a cue combination model (Wei & Körding, 2009). The model and experiments demonstrate the role of visual uncertainty about error feedback in implicit adaptation.In the introduction, the authors notice that implicit adaptation (as measured in error-clamp based paradigms) does not saturate at larger perturbations, but decreases again (e.g. Moorehead et al., 2017 shows no adaptation at 135{degree sign} and 175{degree sign} perturbations). They hypothesized that visual uncertainty about cursor position increases with larger perturbations since the cursor is further from the fixated target. This could decrease importance assigned to visual feedback which could explain lower asymptotes.The authors characterize visual uncertainty for 3 rotation sizes in a first experiment, and while this experiment could be improved, it is probably sufficient for the current purposes. Then the authors present a second experiment where adaptation to 7 clamped errors are tested in different groups of participants. The models' visual uncertainty is set using a linear fit to the results from experiment 1, and the remaining 4 parameters are then fit to this second data set. The 4 parameters are (1) proprioceptive uncertainty, (2) uncertainty about the predicted hand position, (3) a learning rate and (4) a retention rate. The authors' Perceptual Error Adaptation model ("PEA") predicts asymptotic levels of implicit adaptation much better than both the PReMo model (Tsay, Kim et al., 2022), which predicts saturated asymptotes, or a causal inference model (Wei & Körding, 2007) which predicts no adaptation for larger rotations. In a third experiment, the authors test their model's predictions about proprioceptive recalibration, but unfortunately compare their data with an unsuitable other data set (Tsay et al. 2020, instead of Tsay et al. 2021). Finally, the authors conduct a fourth experiment where they put their model to the test. They measure implicit adaptation with increased visual uncertainty, by adding blur to the cursor, and the results are again better in line with their model (predicting overall lower adaptation), than with the PReMo model (predicting equal saturation but at larger perturbations) or a causal inference model (predicting equal peak adaptation, but shifted to larger rotations). In particular the model fits for experiment 2 and the results from experiment 4 show that the core idea of the model has merit: increased visual uncertainty about errors dampens implicit adaptation.(2.2) StrengthsIn this study the authors propose a Perceptual Error Adaptation model ("PEA") and the work combines various ideas from the field of cue combination, Bayesian methods and new data sets, collected in four experiments using various techniques that test very different components of the model. The central component of visual uncertainty is assessed in a first experiment. The model uses 4 other parameters to explain implicit adaptation. These parameters are: (1) a learning and (2) a retention rate, as used in popular state space models and the uncertainty (variance) of (3) predicted and (4) proprioceptive hand position. In particular, the authors observe that asymptotes for implicit learning do not saturate, as claimed before, but decrease again when rotations are very large and that this may have to do with visual uncertainty (e.g. Tsay et al., 2021, J Neurophysiol 125, 12-22). The final experiment confirms predictions of the fitted model about what happens when visual uncertainty is increased (overall decrease of adaptation). By incorporating visual uncertainty depending on retinal eccentricity, the predictions of the PEA model for very large perturbations are notably different from, and better than, the predictions of the two other models it is compared to. That is, the paper provides strong support for the idea that visual uncertainty of errors matters for implicit adaptation.(2.3) WeaknessesAlthough the authors don't say this, the "concave" function that shows that adaptation does not saturate for larger rotations has been shown before, including in papers cited in this manuscript.

For a proper citation of the “concave” adaptation function: we assume the reviewer is referring to the study by Morehead, 2017 which tested large clamp sizes up to 135 o and 175 o. Unsurprisingly, the 135 o and 175 o conditions lead to nearly zero adaptation, possibly due to the trivial fact that people cannot even see the moving cursor. We have quoted this seminar study from the very beginning. All other error-clamp studies with a block design emphasized an invariant or saturated implicit adaptation with large rotations (e.g., Kim, et al., 2019).

The first experiment, measuring visual uncertainty for several rotation sizes in error-clamped paradigms has several shortcomings, but these might not be so large as to invalidate the model or the findings in the rest of the manuscript. There are two main issues we highlight here. First, the data is not presented in units that allow comparison with vision science literature. Second, the 1 second delay between movement endpoint and disappearance of the cursor, and the presentation of the reference marker, may have led to substantial degradation of the visual memory of the cursor endpoint. That is, the experiment could be overestimating the visual uncertainty during implicit adaptation.

For the issues related to visual uncertainty measurement in Exp1:

First, our visual uncertainty is about cursor motion direction in the display plane, and the measurement in Exp1 has never been done before. Thus, we do not think our data is comparable to any findings in visual science about fovea/peripheral comparison. We quoted Klein and others’ work (Klein & Levi, 1987; Levi et al., 1987) in vision science since their studies showed that the deviation from the fixation is associated with an increase in visual uncertainty. Their study thus inspired us to conduct Exp1 to probe how our concerned visual uncertainty (specifically for visual motion direction) changes with an increasing deviation from the fixation. Any model and its model parameters should be specifically tailored to the task or context it tries to emulate. In our case, motion direction in a center-out-reaching setting is the modeled context, and all the relevant model parameters should be specified in movement angles. This is particularly important since we need to estimate parameters from one experiment to predict behaviors in another experiment.

Second, the 1s delay of the reference cursor has minimal impact on the estimate of visual uncertainty based on previous vision studies. Our Exp1 used a similar visual paradigm by (White et al., 1992), which shows that delay does not lead to an increase in visual uncertainty over a broad range of values (from 0.2s to >1s, see their Figure 5-6).

These two problems have been addressed in the revised manuscript, with proper citations listed.

The paper's third experiment relies to a large degree on reproducing patterns found in one particular paper, where the reported hand positions - as a measure of proprioceptive sense of hand position - are given and plotted relative to an ever present visual target, rather than relative to the actual hand position. That is, (1) since participants actively move to a visual target, the reported hand positions do not reflect proprioception, but mostly the remembered position of the target participants were trying to move to, and (2) if the reports are converted to a difference between the real and reported hand position (rather than the difference between the target and the report), those would be on the order of ~20° which is roughly two times larger than any previously reported proprioceptive recalibration, and an order of magnitude larger than what the authors themselves find (1-2°) and what their model predicts. Experiment 3 is perhaps not crucial to the paper, but it nicely provides support for the idea that proprioceptive recalibration can occur with error-clamped feedback.

Reviewer 3 thinks Tsay 2020 dataset is not appropriate for our theorization, but we respectfully disagree. For the three points raised here, we would like to elaborate:

(1) As we addressed in the previous response, the reported hand location in Figure 4A (Tsay et al., 2020) is not from a test of proprioceptive recalibration as conventionally defined. In the revision, we explicitly state that this dataset is not about proprioceptive recalibration and also delete texts that might mislead people to think so (see Results section). Instead, proprioceptive recalibration is measured by passive movement, as in our Exp3 (Figure 4E). For error-clamp adaptation here, "the remembered position of the target" is the target. Clearly, the participants did not report the target position, which is ever-present. Instead, their reported hand location shows an interestingly continuous change with ongoing adaptation.

(2) Since the Tsay 2020 dataset is not a so-called proprioceptive recalibration, we need not take the difference between the reported location and the actual hand location. Indeed, the difference would be ~20 degrees, but comparing it to the previously reported proprioceptive recalibration is like comparing apples to oranges. In fact, throughout the paper, we refer to the results in Fig 4A as “reported hand location”, not proprioceptive recalibration. The target direction is defined as zero degree thus its presence will not bias the reported hand in the Bayesian cue combination (as this visual cue has a mean value of 0). Using the target as the reference also simplifies our modeling.

(3) Exp3 is crucial for our study since it shows our model and its simple Bayesian cue combination principle are applicable not only to implicit adaptation but also to proprioceptive measures during adaptation. Furthermore, it reproduced the so-called proprioceptive recalibration and explained it away with the same Bayesian cue combination as the adaptation. We noticed that this field has accumulated an array of findings on proprioceptive changes induced by visuomotor adaptation. However, currently, there is a lack of a computational model to quantitatively explain them. Our study at least made an initial endeavor to model these changes.

Perhaps the largest caveat to the study is that it assumes that people do not look at the only error feedback available to them (and can explicitly suppress learning from it). This was probably true in the experiments used in the manuscript, but unlikely to be the case in most of the cited literature. Ignoring errors and suppressing adaptation would also be a disastrous strategy to use in the real world, such that our brains may not be very good at this. So the question remains to what degree - if any - the ideas behind the model generalize to experiments without fixation control, and more importantly, to real life situations.

The largest caveat raised by the reviewer appears to be directed to the error-clamp paradigm in general, not only to our particular study. In essence, this paradigm indeed requires participants to ignore the clamped error; thus, its induced adaptive response can be attributed to implicit adaptation. The original paper that proposed this paradigm (Morehead et al., 2017) has been cited 220 times (According to Google Scholar, at the time of this writing, 06/2024), indicating that the field has viewed this paradigm in a favorable way.

Furthermore, we agree that this kind of instruction and feedback (invariant clamp) differ from daily life experience, but it does not prevent us from gaining theoretical insights by studying human behaviors under this kind of "artificial" task setting. Thinking of the saccadic adaptation (Deubel, 1987; Kojima et al., 2004): jumping the target while the eye moves towards it, and this somewhat artificial manipulation again makes people adapt implicitly, and the adaptation itself is a "disastrous" strategy for real-life situations. However, scientists have gained an enormous understanding of motor adaptation using this seemingly counterproductive adaptation in real life. Also, think of perceptual learning of task-irrelevant stimuli (Seitz & Watanabe, 2005, 2009): when participants are required to learn to discriminate one type of visual stimuli, the background shows another type of stimuli, which people gradually learn even though they do not even notice its presence. This "implicit" learning can be detrimental to our real life, too, but the paradigm itself has advanced our understanding of the inner workings of the cognitive system.

**Recommendations for the authors:**

**Reviewer #2 (Recommendations For The Authors):**
L101: There is a typo: (Tsay et al., 2020, 2020) should be corrected to (Tsay et al., 2020).

Thanks for pointing it out, we corrected this typo.

L224-228: It would be beneficial to evaluate the validity of the estimated sigma_u and sigma_p based on previous reports.

We can roughly estimate *σu* by evaluating the variability of reaching angles during the baseline phase when no perturbation is applied. The standard deviation of the reaching angle in Exp 2 is 5.128o±0.190o, which is close to the *σu* estimated by the model (5.048o). We also used a separate perceptual experiment to test the proprioceptive uncertainty (n = 13, See Figure S6), *σp* from this experiment is 9.737o±5.598o, also close to the *σp* extracted by the model (11.119o). We added these new analysis results to the final version of the paper.

L289-298: I found it difficult to understand the update equations of the proprioceptive calibration based on the PEA model. Providing references to the equations or better explanations would be helpful.

We expanded the process of proprioceptive calibration in Supplementary Text 1 with step-by-step equations and more explanations.

**Reviewer #3 (Recommendations For The Authors):**
Suggestions (or clarification of previous suggestions) for revisionsThe authors persist on using the Tsay et al 2020 paper despite its many drawbacks which the authors attempt to address in their reply. But the main drawback is that the results in the 2020 paper is NOT relative to the unseen hand but to the visual target the participants were supposed to move their hand to. If the results were converted so to be relative to the unseen hand, the localization biases would be over 20 deg in magnitude.The PEA simulations are plotted relative to the unseen hand which makes sense. If the authors want to persist using the Tsay 2020 dataset despite any issues, they at least need to make sure that the simulations are mimicking the same change. That is, the data from Tsay 2020 needs to be converted to the same variable used in the current paper.If the main objection for using the Tsay 2021 is that the design would lead to forgetting, we found that active localization (or any intervening active movements like no-cursor reach) does lead to some interference or forgetting (a small reduction in overall magnitude of adaptation) this is not the case for passive localization, see Ruttle et al, 2021 (data on osf). This was also just a suggestion, there may of course also be other, more suitable data sets.

As stated above, changing the reference system is not necessary, nor does it affect our results. Tsay et al 2020 dataset is unique since it shows the gradual change of reported hand location along with error-clamp adaptation. The forgetting (or reduction in proprioceptive bias), even if it exists, would not affect the fitting quality of our model for the Tsay 2020 dataset: if we assume that forgetting is invariant over the adaptation process, the forgetting would only reduce the proprioceptive bias uniformly across trials. This can be accounted for by a smaller weight on x^hand . The critical fact is that the model can explain the gradual drift of the proprioceptive judgment of the hand location.

By the way, Ruttle et al.'s 2021 dataset is not for error-clamp adaptation, and thus we will leave it to test our model extension in the future (after incorporating an explicit process in the model).

References

Deubel, H. (1987). Adaptivity of gain and direction in oblique saccades. *Eye Movements from Physiology to Cognition*. https://www.sciencedirect.com/science/article/pii/B9780444701138500308

Kim, H. E., Parvin, D. E., & Ivry, R. B. (2019). The influence of task outcome on implicit motor learning. *ELife*, *8*. https://doi.org/10.7554/eLife.39882

Klein, S. A., & Levi, D. M. (1987). Position sense of the peripheral retina. *JOSA A*, *4*(8), 1543–1553.

Kojima, Y., Iwamoto, Y., & Yoshida, K. (2004). Memory of learning facilitates saccadic adaptation in the monkey. *The Journal of Neuroscience: The Official Journal of the Society for Neuroscience*, *24*(34), 7531–7539.

Levi, D. M., Klein, S. A., & Yap, Y. L. (1987). Positional uncertainty in peripheral and amblyopic vision. *Vision Research*, *27*(4), 581–597.

Morehead, J. R., Taylor, J. A., Parvin, D. E., & Ivry, R. B. (2017). Characteristics of implicit sensorimotor adaptation revealed by task-irrelevant clamped feedback. *Journal of Cognitive Neuroscience*, *29*(6), 1061–1074.

Seitz, & Watanabe. (2005). A unified model for perceptual learning. *Trends in Cognitive Sciences*, *9*(7), 329–334.

Seitz, & Watanabe. (2009). The phenomenon of task-irrelevant perceptual learning. *Vision Research*, *49*(21), 2604–2610.

Sivak, B., & Mackenzie, C. L. (1992). Chapter 10 The Contributions of Peripheral Vision and Central Vision to Prehension. In L. Proteau & D. Elliott (Eds.), *Advances in Psychology* (Vol. 85, pp. 233–259). North-Holland.

Tsay, J. S., Avraham, G., Kim, H. E., Parvin, D. E., Wang, Z., & Ivry, R. B. (2021). The effect of visual uncertainty on implicit motor adaptation. *Journal of Neurophysiology*, *125*(1), 12–22.

Tsay, J. S., Kim, H. E., Saxena, A., Parvin, D. E., Verstynen, T., & Ivry, R. B. (2022). Dissociable use-dependent processes for volitional goal-directed reaching. *Proceedings. Biological Sciences / The Royal Society*, *289*(1973), 20220415.

Tsay, J. S., Kim, H., Haith, A. M., & Ivry, R. B. (2022). Understanding implicit sensorimotor adaptation as a process of proprioceptive re-alignment. *ELife*, *11*, e76639.

Tsay, J. S., Parvin, D. E., & Ivry, R. B. (2020). Continuous reports of sensed hand position during sensorimotor adaptation. *Journal of Neurophysiology*, *124*(4), 1122–1130.

Tsay, J. S., Tan, S., Chu, M. A., Ivry, R. B., & Cooper, E. A. (2023). Low Vision Impairs Implicit Sensorimotor Adaptation in Response to Small Errors, But Not Large Errors. *Journal of Cognitive Neuroscience*, *35*(4), 736–748.

White, J. M., Levi, D. M., & Aitsebaomo, A. P. (1992). Spatial localization without visual references. *Vision Research*, *32*(3), 513–526.

The following is the authors’ response to the original reviews.

**eLife assessment**
This study presents a valuable finding on the influence of visual uncertainty and Bayesian cue combination on implicit motor adaptation in young healthy participants. The evidence supporting the claims of the authors is solid, although a better discussion of the link between the model variables and the outcomes of related behavioral experiments would strengthen the conclusions. The work will be of interest to researchers in sensory cue integration and motor learning.
**Public Reviews:**

**Reviewer #1 (Public Review):**
This valuable study demonstrates a novel mechanism by which implicit motor adaptation saturates for large visual errors in a principled normative Bayesian manner. Additionally, the study revealed two notable empirical findings: visual uncertainty increases for larger visual errors in the periphery, and proprioceptive shifts/implicit motor adaptation are non-monotonic, rather than ramp-like. This study is highly relevant for researchers in sensory cue integration and motor learning. However, I find some areas where statistical quantification is incomplete, and the contextualization of previous studies to be puzzling.

Thank you for your feedback and the positive highlights of our study. We appreciate your insights and will address the concerns in our revisions.

Issue #1: Contextualization of past studies.While I agree that previous studies have focused on how sensory errors drive motor adaptation (e.g., Burge et al., 2008; Wei and Kording, 2009), I don't think the PReMo model was contextualized properly. Indeed, while PReMo should have adopted clearer language - given that proprioception (sensory) and kinaesthesia (perception) have been used interchangeably, something we now make clear in our new study (Tsay, Chandy, et al. 2023) - PReMo's central contribution is that a perceptual error drives implicit adaptation (see Abstract): the mismatch between the felt (perceived) and desired hand position. The current paper overlooks this contribution. I encourage the authors to contextualize PReMo's contribution more clearly throughout. Not mentioned in the current study, for example, PReMo accounts for the continuous changes in perceived hand position in Figure 4 (Figure 7 in the PReMo study).There is no doubt that the current study provides important additional constraints on what determines perceived hand position: Firstly, it offers a normative Bayesian perspective in determining perceived hand position. PReMo suggests that perceived hand position is determined by integrating motor predictions with proprioception, then adding a proprioceptive shift; PEA formulates this as the optimal integration of these three inputs. Secondly, PReMo assumed visual uncertainty to remain constant for different visual errors; PEA suggests that visual uncertainty ought to increase (but see Issue #2).

Thank you for the comments and suggestions. We have now incorporated the citation for (Tsay et al., 2024), to acknowledge their clarification on the terms of perceptual error. We also agree that our model differs in two fundamental ways. One is to ditch the concept of proprioceptive shift and its contribution to the perceived hand location; instead, we resort to a “one-shot” integration of three types of cues with Bayesian rules. This is a more elegant and probably more ecological way of processing hand location per Occam's Razor. The second essential change is to incorporate the dependency of visual uncertainty on perturbation size into the model, as opposed to resorting to a ramp function of proprioceptive changes relative to perturbation size. The ramp function is not well grounded in perception studies. Yes, we acknowledged that PReMo is the first to recognize the importance of perceptual error, but highlighted the model differences in our Discussion.

We also think the PReMo model has the potential to explain Fig 4A. But the Tsay et al., 2022 paper assumes that “a generic shift in visual space” explains the gradual proprioceptive changes from negative to positive (see page 17 in Tsay et al., 2022). We do not think that evoking this visual mechanism is necessary to explain Fig 4A; instead, the proprioceptive change is a natural result of hand deviations during implicit adaptation. As the hand moves away from the target (in the positive direction) during adaptation, the estimated hand location goes alone with it. We believe this is the correct way of explaining Fig4A results. As we played around with the PReMo model, we found it is hard to use visual shift to explain this part of data without additional assumptions (at least not with the ones published in Tsay et al., 2022). Furthermore, our PEA model also parsimoniously explains away the proprioceptive shift observed in a completely different setting, i,e., the proprioceptive changes measured by the passive method as a function of perturbation size in Exp 3.

We expanded the discussion about the comparison between the two models, especially about their different views for explaining Fig4A.

Issue #2: Failed replication of previous results on the effect of visual uncertainty.(2a) A key finding of this paper is that visual uncertainty linearly increases in the periphery; a constraint crucial for explaining the non-monotonicity in implicit adaptation. One notable methodological deviation from previous studies is the requirement to fixate on the target: Notably, in the current experiments, participants were asked to fixate on the target, a constraint not imposed in previous studies. In a free-viewing environment, visual uncertainty may not attenuate as fast, and hence, implicit adaptation does not attenuate as quickly as that revealed in the current design with larger visual errors. Seems like this current fixation design, while important, needs to be properly contextualized considering how it may not represent most implicit adaptation experiments.

First, we don’t think there is any previous study that examined visual uncertainty as a function of perturbation size. Thus, we do not have a replication problem here. Secondly, our data indicate that even without asking people to fixate on the target, people still predominantly fixate on the target during error-clamp adaptation (when they are “free” viewing). For our Exp 1, the fixation on the straight line between the starting position and the target is 86%-95% (as shown in Figure S1 now， also see below). We also collected eye-tracking data in Exp 4, which is a typical error-clamp experiment. More than 95% fall with +/- 50 pixels around the center of the screen, even slightly higher than Exp 1. This is well understandable: the typical error-clamp adaptation requires people to ignore the cursor and move the hand towards the target. To minimize the interference of the concurrently moving cursor, people depend on the fixation on the target, the sole task-relevant visual marker in the workspace, to achieve the task goal.

In sum, forcing the participants to fixate on the target is not because we aimed to make up the linear dependency of visual uncertainty; we required them to do so to mimic the eye-tracking pattern in typical error-clamp learning, which has been revealed in our pilot experiment. The visual uncertainty effect is sound, our study is the first to clearly demonstrate it.

**Author response image 1. sa4fig1:** 

On a side note (but an important one), the high percentage of fixation on the aiming target is also true for conventional visuomotor rotation, which involves strategic re-aiming (shown in Bromberg et al., 2019; de Brouwer et al., 2018, we have an upcoming paper to show this). This is one reason that our new theory would also be applicable to other types of motor adaptation.

(2b) Moreover, the current results - visual uncertainty attenuates implicit adaptation in response to large, but not small, visual errors - deviates from several past studies that have shown that visual uncertainty attenuates implicit adaptation to small, but not large, visual errors (Tsay, Avraham, et al. 2021; Makino, Hayashi, and Nozaki, n.d.; Shyr and Joshi 2023). What do the authors attribute this empirical difference to? Would this free-viewing environment also result in the opposite pattern in the effect of visual uncertainty on implicit adaptation for small and large visual errors?

We don’t think all the mentioned previous studies manipulated the visual uncertainty in a parametric way, and none of them provided quantitative measures of visual uncertainty. As we detailed in our Exp4 and in our Discussion, we don’t think Tsay et al., 2021 paper’s manipulation of visual uncertainty is appropriate (see below for 2d). Makino et al., 2023 study used multiple clamped cursors to perturb people, and its effect is not easily accountable since additional processes might be invoked given this kind of complex visual feedback. More importantly, we do not think this is a direct way of modulating visual uncertainty, nor did they provide any evidence.

(2c) In the current study, the measure of visual uncertainty might be inflated by brief presentation times of comparison and referent visual stimuli (only 150 ms; our previous study allowed for a 500 ms viewing time to make sure participants see the comparison stimuli). Relatedly, there are some individuals whose visual uncertainty is greater than 20 degrees standard deviation. This seems very large, and less likely in a free-viewing environment.

For our 2AFC, the reference stimulus is the actual clamped cursor, which lasts for 800 ms. The comparison stimulus is a 150-ms dot representation appearing near the reference. For measuring perception of visual motion, this duration is sufficient as previous studies used similar durations (Egly & Homa, 1984; Owsley et al., 1995). We think the 20-degree standard deviation is reasonable given that people fixate on the target, with only peripheral vision to process the fast moving cursor. The steep linear increase in visual uncertainty about visual motion is well documented. The last author of this paper has shown that the uncertainty of visual motion speed (though not about angels) follows the same steep trend (Wei et al., 2010). It is noteworthy that without using our measured visual uncertainty in Exp1, if we fit the adaptation data in Exp2 to “estimate” the visual uncertainty, they are in fact well aligned with each other (see Figure S7 and Supplementary Text 2). This is a strong support that our estimation is valid and accurate. We think this high visual uncertainty is an important message to the field. Thus we now highlighted its magnitude in our Discussion.

(2d) One important confound between clear and uncertain (blurred) visual conditions is the number of cursors on the screen. The number of cursors may have an attenuating effect on implicit adaptation simply due to task-irrelevant attentional demands (Parvin et al. 2022), rather than that of visual uncertainty. Could the authors provide a figure showing these blurred stimuli (gaussian clouds) in the context of the experimental paradigm? Note that we addressed this confound in the past by comparing participants with and without low vision, where only one visual cursor is provided for both groups (Tsay, Tan, et al. 2023).

Thank you for raising this important point about types of visual stimuli for manipulating uncertainty. We used Gaussian blur of a single cursor (similar to Burge et al., 2008) instead of a cloud of dots. We now added a figure inset to show how this blur looks.

Using a cursor cloud Makino et al., 2023; Tsay et al., 2021 to modulate visual uncertainty has inherent drawbacks that make it unsuitable for visuomotor adaptation. For the error clamp paradigm, the error is defined as angular deviation. The cursor cloud consists of multiple cursors spanning over a range of angles, which affects both the sensory uncertainty (the intended outcome) and the sensory estimate of angles (the error estimate, the undesired outcome). In Bayesian terms, the cursor cloud aims to modulate the sigma of a distribution (sigma_v in our model), but it additionally affects the mean of the distribution (mu). This unnecessary confound is avoided by using cursor blurring, which is still a cursor with its center (mu) unchanged from a single cursor. Furthermore, as correctly pointed out in the original paper by Tsay et al., 2021, the cursor cloud often overlaps with the visual target, this “target hit” would affect adaptation, possibly via a reward learning mechanism (See Kim et al., 2019). This is a second confound that accompanies the cursor cloud.

Issue #3: More methodological details are needed.(3a) It's unclear why, in Figure 4, PEA predicts an overshoot in terms of perceived hand position from the target. In PReMo, we specified a visual shift in the perceived target position, shifted towards the adapted hand position, which may result in overshooting of the perceived hand position with this target position. This visual shift phenomenon has been discovered in previous studies (e.g., (Simani, McGuire, and Sabes 2007)).

Visual shift, as it is called in Simani et al., 2007, is irrelevant for our task here. The data we are modeling are motor adaptation (hand position changes) and so-called proprioceptive changes (hand localization changes), both are measured and referenced in the extrinsic coordinate, not referenced to a visual target. For instance, the proprioceptive changes are either relative to the actual hand location (Exp 3) or relative to the goal (Fig 4A). We also don’t think visual shift is necessary in explaining the perceptual judgment of an unseen hand (the target shown during the judgment indeed has an effect of reducing the biasing effect of PE, see below for responses to reviewer 3).

In the PEA model, the reported hand angle is the result of integrating cues from the actual hand position and the estimated hand position (x_hand_hat) from previous movements. This integration process leads to the combined reported hand position potentially overshooting or undershooting, depending on the degree of adaptation. It is the changed proprioceptive cue (because the actively moved hand slowly adapted to the error clamp) leading to the overshoot of the perceived hand position.

In Results, we now explain these value changes with parentheses. Model details about the mechanisms of cue combination and model predictions can be found in Supplementary Text 1. We believe these detailed explanations can make this apparent.

(3b) The extent of implicit adaptation in Experiment 2, especially with smaller errors, is unclear. The implicit adaptation function seems to be still increasing, at least by visual inspection. Can the authors comment on this trend, and relatedly, show individual data points that help the reader appreciate the variability inherent to these data?

Indeed, the adaptation for small errors appears not completely saturated with our designated number of trials. However, this will not affect our model analysis. Our model fitting for PEA and other competing models is done on the time-series of adaptation, not on the saturated adaptation extent (see Fig 3A). Thus, despite that some conditions might not produce the full range of adaptation, the data is sufficient to constrain the models. We now mention this concern in Results; we also emphasize that the model not only explains the adaptation magnitude (operationally defined as adaptation extent measured at the same time, i.e., the end of the adaptation phase) but also the full learning process.

In response, we have included individual data points in the revised Figure 3B-D to provide a clear illustration of the extent of implicit adaptation, particularly for small perturbations.

(3c) The same participants were asked to return for multiple days/experiments. Given that the authors acknowledge potential session effects, with attenuation upon re-exposure to the same rotation (Avraham et al. 2021), how does re-exposure affect the current results? Could the authors provide clarity, perhaps a table, to show shared participants between experiments and provide evidence showing how session order may not be impacting results?

Thank you for raising the issue of session and re-exposure effects. First, we don’t think Exp1 has an effect on Exp4. Exp1 is a perceptual task and Exp4 is a motor adaptation task. Furthermore, Exp1 used random visual stimuli on both sides, thus it did not lead to any adaptation effect on its own. Second, Exp4 indeed had three sessions performed on three days, but the session effect does not change our main conclusion about the visual uncertainty. We used a 3-way repeated-measures anova (3 day x 3 perturbation x 2 visual uncertainty) revealed a significant main effect of day (F(2,36) = 17.693, p<0.001), indicating changes in performance across sessions (see Figure below). Importantly, the effects of perturbation and visual uncertainty (including their interactions) remain the same. The day factor did not interact with them. The main effect of day shows that the overall adaptation effect is reduced across days. Post-hoc pairwise comparisons elucidated that single-trial learning (STL) performance on Day 1 was significantly higher than on Day 2 (p = 0.004) and Day 3 (p < 0.001), with no significant difference between Day 2 and Day 3 (p = 0.106). Other ANOVA details: significant main effects for perturbation (F(1,36) = 8.872, p<0.001) and visual uncertainty (F(1,18) = 49.164, p<0.001), as well as a significant interaction between perturbation size and visual uncertainty (F(2,36) = 5.160, p = 0.013). There were no significant interactions involving the day factor with any other factors (all p > 0.182). Thus, the overall adaptation decreases over the days, but the day does not affect our concerned interaction effect of visual uncertainty and perturbation. The fact that their interaction preserved over different sessions strengthened our conclusion about how visual uncertainty systematically affects implicit adaptation.

**Author response image 2. sa4fig2:** 

(3d) The number of trials per experiment should be detailed more clearly in the Methods section (e.g., Exp 4). Moreover, could the authors please provide relevant code on how they implemented their computational models? This would aid in future implementation of these models in future work. I, for one, am enthusiastic to build on PEA.

We have clarified the number of trials conducted in each experiment, with detailed information now readily available in the Methods section of the main text. In addition, we have made the code for data analysis and modeling publicly accessible. These resources can be found in the updated "Data Availability" section of our paper.

(3f) In addition to predicting a correlation between proprioceptive shift and implicit adaptation on a group level, both PReMo and PEA (but not causal inference) predict a correlation between individual differences in proprioceptive shift and proprioceptive uncertainty with the extent of implicit adaptation (Tsay, Kim, et al. 2021). Interestingly, shift and uncertainty are independent (see Figures 4F and 6C in Tsay et al, 2021). Does PEA also predict independence between shift and uncertainty? It seems like PEA does predict a correlation.

Thank you for addressing this insightful question. Our PEA model indeed predicts a positive correlation (although not linear) between the proprioceptive uncertainty and the amplitude of the estimated hand position (x_hand_hat). This prediction is consistent with the simulations conducted, using the same parameters that were applied to generate the results depicted in

Figure 4B of our manuscript (there is a sign flip as x_hand_hat is negative).

**Author response image 3. sa4fig3:** 

Regarding the absence of a correlation observed in Tsay et al., 2021, we offer several potential explanations for this discrepancy. First, the variability observed in passive hand localization during motor adaptation (as in Tsay et al., 2021) does not directly equal proprioceptive uncertainty, which typically requires psychophysical testing to accurately assess. Second, our study showed that the proprioceptive bias attenuates during the repetitive measurements; in our Exp3, it decreased within a block of three trials. We noticed that Tsay et al., 2021 study used 36 measurements in a row without interleaving adaptation trials. Thus, the “averaged” proprioceptive bias in Tsay’s study might not reflect the actual bias during adaptation. We also noticed that that study showed large individual differences in both proprioceptive bias and proprioceptive variability (not uncertainty), thus getting a positive result, if it were really there, would require a large number of participants, probably larger than their n=30ish sample size. These putative explanations are not put in the revision, which already has a long discussion and has no space for discussing about a null result.

**Reviewer #2 (Public Review):**
Summary:The authors present the Perceptual Error Adaptation (PEA) model, a computational approach offering a unified explanation for behavioral results that are inconsistent with standard state-space models. Beginning with the conventional state-space framework, the paper introduces two innovative concepts. Firstly, errors are calculated based on the perceived hand position, determined through Bayesian integration of visual, proprioceptive, and predictive cues. Secondly, the model accounts for the eccentricity of vision, proposing that the uncertainty of cursor position increases with distance from the fixation point. This elegantly simple model, with minimal free parameters, effectively explains the observed plateau in motor adaptation under the implicit motor adaptation paradigm using the error-clamp method. Furthermore, the authors experimentally manipulate visual cursor uncertainty, a method established in visuomotor studies, to provide causal evidence. Their results show that the adaptation rate correlates with perturbation sizes and visual noise, uniquely explained by the PEA model and not by previous models. Therefore, the study convincingly demonstrates that implicit motor adaptation is a process of Bayesian cue integrationStrengths:In the past decade, numerous perplexing results in visuomotor rotation tasks have questioned their underlying mechanisms. Prior models have individually addressed aspects like aiming strategies, motor adaptation plateaus, and sensory recalibration effects. However, a unified model encapsulating these phenomena with a simple computational principle was lacking. This paper addresses this gap with a robust Bayesian integration-based model. Its strength lies in two fundamental assumptions: motor adaptation's influenced by visual eccentricity, a well-established vision science concept, and sensory estimation through Bayesian integration. By merging these well-founded principles, the authors elucidate previously incongruent and diverse results with an error-based update model. The incorporation of cursor feedback noise manipulation provides causal evidence for their model. The use of eye-tracking in their experimental design, and the analysis of adaptation studies based on estimated eccentricity, are particularly elegant. This paper makes a significant contribution to visuomotor learning research.Weaknesses:The paper provides a comprehensive account of visuomotor rotation paradigms, addressing incongruent behavioral results with a solid Bayesian integration model. However, its focus is narrowly confined to visuomotor rotation, leaving its applicability to broader motor learning paradigms, such as force field adaptation, saccadic adaptation, and de novo learning paradigms, uncertain. The paper's impact on the broader fields of neuroscience and cognitive science may be limited due to this specificity. While the paper excellently demonstrates that specific behavioral results in visuomotor rotation can be explained by Bayesian integration, a general computational principle, its contributions to other motor learning paradigms remain to be explored. The paper would benefit from a discussion on the model's generality and its limitations, particularly in relation to the undercompensating effects in other motor learning paradigms.

Thank you for your thoughtful review and recognition of the contributions our work makes towards understanding implicit motor adaptation through the Perceptual Error Adaptation (PEA) model. We appreciate your suggestion to broaden the discussion about the model's applicability beyond the visuomotor rotation paradigm, a point we acknowledge was not sufficiently explored in our initial discussion.

Our model is not limited to the error-clamp adaptation, where the participants were explicitly told to ignore the rotated cursor. The error-clamp paradigm is one rare example that implicit motor learning can be isolated in a nearly idealistic way. Our findings thus imply two key aspects of implicit adaptation: (1) localizing one’s effector is implicitly processed and continuously used to update the motor plan; (2) Bayesian cue combination is at the core of integrating movement feedback and motor-related cues (motor prediction cue in our model) when forming procedural knowledge for action control.

We will propose that the same two principles should be applied to various kinds of motor adaptation and motor skill learning, which constitutes motor learning in general. Most of our knowledge about motor adaptation is from visuomotor rotation, prism adaptation, force field adaptation, and saccadic adaptation. The first three types all involve localizing one’s effector under the influence of perturbed sensory feedback, and they also have implicit learning. We believe they can be modeled by variants of our model, or at least should consider using the two principles we laid out above to think of their computational nature. For skill learning, especially for de novo learning, the area still lacks a fundamental computational model that accounts for skill acquisition process on the level of relevant movement cues. Our model suggests a promising route, i.e., repetitive movements with a Bayesian cue combination of movement-related cues might underlie the implicit process of motor skills.

We added more discussion on the possible broad implications of our model in the revision.

**Reviewer #3 (Public Review):**
SummaryIn this paper, the authors model motor adaptation as a Bayesian process that combines visual uncertainty about the error feedback, uncertainty about proprioceptive sense of hand position, and uncertainty of predicted (=planned) hand movement with a learning and retention rate as used in state space models. The model is built with results from several experiments presented in the paper and is compared with the PReMo model (Tsay, Kim, et al., 2022) as well as a cue combination model (Wei & Körding, 2009). The model and experiments demonstrate the role of visual uncertainty about error feedback in implicit adaptation.In the introduction, the authors notice that implicit adaptation (as measured in error-clamp-based paradigms) does not saturate at larger perturbations, but decreases again (e.g. Moorehead et al., 2017 shows no adaptation at 135{degree sign} and 175{degree sign} perturbations). They hypothesized that visual uncertainty about cursor position increases with larger perturbations since the cursor is further from the fixated target. This could decrease the importance assigned to visual feedback which could explain lower asymptotes.The authors characterize visual uncertainty for 3 rotation sizes in the first experiment, and while this experiment could be improved, it is probably sufficient for the current purposes. Then the authors present a second experiment where adaptation to 7 clamped errors is tested in different groups of participants. The models' visual uncertainty is set using a linear fit to the results from experiment 1, and the remaining 4 parameters are then fit to this second data set. The 4 parameters are (1) proprioceptive uncertainty, (2) uncertainty about the predicted hand position, (3) a learning rate, and (4) a retention rate. The authors' Perceptual Error Adaptation model ("PEA") predicts asymptotic levels of implicit adaptation much better than both the PReMo model (Tsay, Kim et al., 2022), which predicts saturated asymptotes, or a causal inference model (Wei & Körding, 2007) which predicts no adaptation for larger rotations. In a third experiment, the authors test their model's predictions about proprioceptive recalibration, but unfortunately, compare their data with an unsuitable other data set. Finally, the authors conduct a fourth experiment where they put their model to the test. They measure implicit adaptation with increased visual uncertainty, by adding blur to the cursor, and the results are again better in line with their model (predicting overall lower adaptation) than with the PReMo model (predicting equal saturation but at larger perturbations) or a causal inference model (predicting equal peak adaptation, but shifted to larger rotations). In particular, the model fits experiment 2 and the results from experiment 4 show that the core idea of the model has merit: increased visual uncertainty about errors dampens implicit adaptation.StrengthsIn this study, the authors propose a Perceptual Error Adaptation model ("PEA") and the work combines various ideas from the field of cue combination, Bayesian methods, and new data sets, collected in four experiments using various techniques that test very different components of the model. The central component of visual uncertainty is assessed in the first experiment. The model uses 4 other parameters to explain implicit adaptation. These parameters are (1) learning and (2) retention rate, as used in popular state space models, and the uncertainty (variance) of (3) predicted and (4) proprioceptive hand position. In particular, the authors observe that asymptotes for implicit learning do not saturate, as claimed before, but decrease again when rotations are very large and that this may have to do with visual uncertainty (e.g. Tsay et al., 2021, J Neurophysiol 125, 12-22). The final experiment confirms predictions of the fitted model about what happens when visual uncertainty is increased (overall decrease of adaptation). By incorporating visual uncertainty depending on retinal eccentricity, the predictions of the PEA model for very large perturbations are notably different from and better than, the predictions of the two other models it is compared to. That is, the paper provides strong support for the idea that visual uncertainty of errors matters for implicit adaptation.WeaknessesAlthough the authors don't say this, the "concave" function that shows that adaptation does not saturate for larger rotations has been shown before, including in papers cited in this manuscript.The first experiment, measuring visual uncertainty for several rotation sizes in error-clamped paradigms has several shortcomings, but these might not be so large as to invalidate the model or the findings in the rest of the manuscript. There are two main issues we highlight here. First, the data is not presented in units that allow comparison with vision science literature. Second, the 1 second delay between the movement endpoint and the disappearance of the cursor, and the presentation of the reference marker, may have led to substantial degradation of the visual memory of the cursor endpoint. That is, the experiment could be overestimating the visual uncertainty during implicit adaptation.The paper's third experiment relies to a large degree on reproducing patterns found in one particular paper, where the reported hand positions - as a measure of proprioceptive sense of hand position - are given and plotted relative to an ever-present visual target, rather than relative to the actual hand position. That is, (1) since participants actively move to a visual target, the reported hand positions do not reflect proprioception, but mostly the remembered position of the target participants were trying to move to, and (2) if the reports are converted to a difference between the real and reported hand position (rather than the difference between the target and the report), those would be on the order of ~20{degree sign} which is roughly two times larger than any previously reported proprioceptive recalibration, and an order of magnitude larger than what the authors themselves find (1-2{degree sign}) and what their model predicts. Experiment 3 is perhaps not crucial to the paper, but it nicely provides support for the idea that proprioceptive recalibration can occur with error-clamped feedback.Perhaps the largest caveat to the study is that it assumes that people do not look at the only error feedback available to them (and can explicitly suppress learning from it). This was probably true in the experiments used in the manuscript, but unlikely to be the case in most of the cited literature. Ignoring errors and suppressing adaptation would also be a disastrous strategy to use in the real world, such that our brains may not be very good at this. So the question remains to what degree - if any - the ideas behind the model generalize to experiments without fixation control, and more importantly, to real-life situations.Specific comments:A small part of the manuscript relies on replicating or modeling the proprioceptive recalibration in a study we think does NOT measure proprioceptive recalibration (Tsay, Parvin & Ivry, JNP, 2020). In this study, participants reached for a visual target with a clamped cursor, and at the end of the reach were asked to indicate where they thought their hand was. The responses fell very close to the visual target both before and after the perturbation was introduced. This means that the difference between the actual hand position, and the reported/felt hand position gets very large as soon as the perturbation is introduced. That is, proprioceptive recalibration would necessarily have roughly the same magnitude as the adaptation displayed by participants. That would be several times larger than those found in studies where proprioceptive recalibration is measured without a visual anchor. The data is plotted in a way that makes it seem like the proprioceptive recalibration is very small, as they plot the responses relative to the visual target, and not the discrepancy between the actual and reported hand position. It seems to us that this study mostly measures short-term visual memory (of the target location). What is astounding about this study is that the responses change over time to begin with, even if only by a tiny amount. Perhaps this indicates some malleability of the visual system, but it is hard to say for sure.Regardless, the results of that study do not form a solid basis for the current work and they should be removed. We would recommend making use of the dataset from the same authors, who improved their methods for measuring proprioception shifts just a year later (Tsay, Kim, Parvin, Stover, and Ivry, JNP, 2021). Although here the proprioceptive shifts during error-clamp adaptation (Exp 2) were tiny, and not quite significant (p<0.08), the reports are relative to the actual location of the passively placed unseen hand, measured in trials separate from those with reach adaptation and therefore there is no visual target to anchor their estimates to.Experiment 1 measures visual uncertainty with increased rotation size. The authors cite relevant work on this topic (Levi & Klein etc) which has found a linear increase in uncertainty of the position of more and more eccentrically displayed stimuli.First, this is a question where the reported stimuli and effects could greatly benefit from comparisons with the literature in vision science, and the results might even inform it. In order for that to happen, the units for the reported stimuli and effects should (also) be degrees of visual angle (dva).As far as we know, all previous work has investigated static stimuli, where with moving stimuli, position information from several parts of the visual field are likely integrated over time in a final estimate of position at the end of the trajectory (a Kalman filter type process perhaps). As far as we know, there are no studies in vision science on the uncertainty of the endpoint of moving stimuli. So we think that the experiment is necessary for this study, but there are some areas where it could be improved.Then, the linear fit is done in the space of the rotation size, but not in the space of eccentricity relative to fixation, and these do not necessarily map onto each other linearly. If we assume that the eye-tracker and the screen were at the closest distance the manufacturer reports it to work accurately at (45 cm), we would get the largest distances the endpoints are away from fixation in dva. Based on that assumed distance between the participant and monitor, we converted the rotation angles to distances between fixation and the cursor endpoint in degrees visual angle: 0.88, 3.5, and 13.25 dva (ignoring screen curvature, or the absence of it). The ratio between the perturbation angle and retinal distance to the endpoint is roughly 0.221, 0.221, and 0.207 if the minimum distance is indeed used - which is probably fine in this case. But still, it would be better to do fit in the relevant perceptual coordinate system.The first distance (4 deg rotation; 0.88 dva offset between fixation and stimulus) is so close to fixation (even at the assumed shortest distance between eye and screen) that it can be considered foveal and falls within the range of noise of eye-trackers + that of the eye for fixating. There should be no uncertainty on or that close to the fovea. The variability in the data is likely just measurement noise. This also means that a linear fit will almost always go through this point, somewhat skewing the results toward linearity. The advantage is that the estimate of the intercept (measurement noise) is going to be very good. Unfortunately, there are only 2 other points measured, which (if used without the closest point) will always support a linear fit. Therefore, the experiment does not seem suitable to test linearity, only to characterize it, which might be sufficient for the current purposes. We'd understand if the effort to do a test of linearity using many more rotations requires too much effort. But then it should be made much clearer that the experiment assumes linearity and only serves to characterize the assumed linearity.Final comment after the consultation session:There were a lot of discussions about the actual interpretation of the behavioral data from this paper with regards to past papers (Tsay et al. 2020 or 2021), and how it matches the different variables of the model. The data from Tsay 2020 combined both proprioceptive information (Xp) and prediction about hand position (Xu) because it involves active movements. On the other hand, Tsay et al. 2021 is based on passive movements and could provide a better measure of Xp alone. We would encourage you to clarify how each of the variables used in the model is mapped onto the outcomes of the cited behavioral experiments.The reviewers discussed this point extensively during the consultation process. The results reported in the Tsay 2020 study reflect both proprioception and prediction. However, having a visual target contributes more than just prediction, it is likely an anchor in the workspace that draws the response to it. Such that the report is dominated by short-term visual memory of the target (which is not part of the model). However, in the current Exp 3, as in most other work investigating proprioception, this is calculated relative to the actual direction.The solution is fairly simple. In Experiment 3 in the current study, Xp is measured relative to the hand without any visual anchors drawing responses, and this is also consistent with the reference used in the Tsay et al 2021 study and from many studies in the lab of D. Henriques (none of which also have any visual reach target when measuring proprioceptive estimates). So we suggest using a different data set that also measures Xp without any other influences, such as the data from Tsay et al 2021 instead.These issues with the data are not superficial and can not be solved within the model. Data with correctly measured biases (relative to the hand) that are not dominated by irrelevant visual attractors would actually be informative about the validity of the PEA model. Dr. Tsay has so much other that we recommend using a more to-the-point data set that could actually validate the PEA model.

As the comments are repetitive at some places, we summarize them into three questions and address it one by one below:

(1) Methodological Concerns about visual uncertainty estimation in Experiment 1: (a) the visual uncertainty is measured in movement angles (degrees), while the unit in vision science is in visual angles (vda). This mismatch of unit hinders direct comparison between the found visual uncertainty and those reported in the literature, and (b) a 1-second delay between movement endpoint and the reference marker presentation causes an overestimate of visual uncertainty due to potential degradation of visual memory. (c) The linear function of visual uncertainty is a result of having only three perturbation sizes.

a) As noted by the reviewer, our visual uncertainty is about cursor motion direction in the display plane, which has never been measured before. We do not think our data is comparable to any findings in visual science about fovea/peripheral comparison. We quoted Klein and others’ work Klein & Levi, 1987; Levi et al., 1987 in vision science since their studies showed that the deviation from the fixation is associated with the increase in visual uncertainty. Their study thus inspired our Exp1 to probe how our concerned visual uncertainty (specifically for visual motion direction) changes with an increasing deviation from the fixation. We believe that any model and its model parameters should be specifically tailored to the task or context it tries to emulate. In our case, motion direction in a center-out reaching setting is the modeled context, and all the relevant model parameters should be specified in movement angles.

b) The 1s delay of the reference cursor appears to have minimum impact on the estimate of visual uncertainty, based on previous vision studies. Our Exp1 used a similar visual paradigm by White et al., 1992, which shows that delay does not lead to an increase in visual uncertainty over a broad range of values (from 0.2s to >1s, see their Figure 5-6). We will add more methodology justifications in our revision.

c) We agree that if more angles are tested we can be more confident about the linearity of visual uncertainty. However, the linear function is a good approximation of visual uncertainty (as shown in Figure 2C). More importantly, our model performance does not hinge on a strict linear function. Say, if it is a power function with an increasing slope, our model will still predict the major findings presented in the paper, as correctly pointed out by the reviewer. It is the increasing trend of visual uncertainty, which is completely overlooked by previous studies, that lead to various seemingly puzzling findings in implicit adaptation. Lastly, without assuming a linear function, we fitted the large dataset of motor adaptation from Exp2 to numerically estimate the visual uncertainty. This estimated visual uncertainty has a strong linear relationship with perturbation size (R = 0.991, p<0.001). In fact, the model-fitted visual uncertainty is very close to the values we obtained in Exp1. We now included this analysis in the revision. See details in Supplementary text 2 and Figure S7.

(2) Experiment 3's: the reviewer argues that the Tsay et al., 2020 data does not accurately measure proprioceptive recalibration, thus it is not suitable for showing our model’s capacity in explaining proprioceptive changes during adaptation.

Response: We agree that the data from Tsay et al., 2020 is not from passive localization, which is regarded as the widely-accepted method to measure proprioceptive recalibration, a recalibration effect in the sensory domain. The active localization, as used in Tsay et al., 2020, is hypothesized as closely related to people’s forward prediction (where people want to go as the reviewer put it in the comments). However, we want to emphasize that we never equated Tsay’s findings as proprioceptive recalibration: throughout the paper we call them “reported hand location”. We reserved “proprioceptive recalibration” to our own Exp3, which used a passive localization method. Thus, we are not guilty of using this term. Secondly, as far as we know, localization bias or changes, no matter measured by passive or active methods, have not been formally modeled quantitatively. We believe our model can explain both, at least in the error-clamp adaptation setting here. Exp3 is for passive localization, the proprioceptive bias is caused by the biasing effect from the just-perceived hand location (X_hand_hat) from the adaptation trial. Tsay et al. 2020 data is for active localization, whose bias shows a characteristic change from negative to positive. This can be explained by just-perceived hand location (X_hand_hat again) and a gradually-adapting hand (X_p). We think this is a significant advance in the realm of proprioceptive changes in adaptation. Of course, our idea can be further tested in other task conditions, e.g., conventional visuomotor rotation or even gain adaptation, which should be left for future studies.

For technical concerns, Tsay et al., 2020 data set is not ideal: when reporting hand location, the participants view the reporting wheel as well as the original target. As correctly pointed out by the reviewer, the presence of the target might provide an anchoring cue for perceptual judgment, which acts as an attractor for localization. If it were the case, our cue combination would predict that this extra attractor effect would lead to a smaller proprioceptive effect than that is currently reported in their paper. The initial negative bias will be closer to the target (zero), and the later positive bias will be closer to the target too. However, the main trend will remain, i.e. the reported hand location would still show the characteristic negative-to-positive change. The attractor effect of the target can be readily modeled by giving less weight to the just-perceived hand location (X_hand_hat). Thus, we would like to keep Tsay et al., 2020 data in our paper but add some explanations of the limitations of this dataset as well as how the model would fare with these limitations.

That being said, our model can explain away both passive and active localization during implicit adaptation elicited by error clamp. The dataset from Tsay et al., 2021 paper is not a good substitute for their 2020 paper in terms of modeling, since that study interleaved some blocks of passive localization trials with adaptation trials. This kind of block design would lead to forgetting of both adaptation (Xp in our model) and the perceived hand (X_hand_hat in our model), the latter is still not considered in our model yet. As our Exp3, which also used passive localization, shows, the influence of the perceived hand on proprioceptive bias is short-lived, up to three trials without adaptation trials. Of course, it would be of great interest to design future studies to study how the proprioceptive bias changes over time, and how its temporal changes relate to the perceptual error. Our model provides a testbed to move forward in this direction.

(3) The reviewer raises concerns about the study's assumption that participants ignore error feedback, questioning the model's applicability to broader contexts and real-world scenarios where ignoring errors might not be viable or common.

Reviewer 2 raised the same question above. We moved our responses here. “We appreciate your suggestion to broaden the discussion about the model's applicability beyond the visuomotor rotation paradigm, a point we acknowledge was not sufficiently explored in our initial discussion.

Our model is not limited to the error-clamp adaptation, where the participants were explicitly told to ignore the rotated cursor. The error-clamp paradigm is one rare example that implicit motor learning can be isolated in a nearly idealistic way. Our findings thus imply two key aspects of implicit adaptation: (1) localizing one’s effector is implicitly processed and continuously used to update the motor plan; (2) Bayesian cue combination is at the core of integrating movement feedback and motor-related cues (motor prediction cue in our model) when forming procedural knowledge for action control.

We will propose that the same two principles should be applied to various kinds of motor adaptation and motor skill learning, which constitutes motor learning in general. Most of our knowledge about motor adaptation is from visuomotor rotation, prism adaptation, force field adaptation, and saccadic adaptation. The first three types all involve localizing one’s effector under the influence of perturbed sensory feedback, and they also have implicit learning. We believe they can be modeled by variants of our model, or at least should consider using the two principles we laid out above to think of their computational nature. For skill learning, especially for de novo learning, the area still lacks a fundamental computational model that accounts for skill acquisition process on the level of relevant movement cues. Our model suggests a promising route, i.e., repetitive movements with a Bayesian cue combination of movement-related cues might underlie the implicit process of motor skills.”

We also add one more important implication of our model: as stated above, our model also explains that the proprioceptive changes, revealed by active or passive localization methods, are brought by (mis)perceived hand localization via Bayesian cue combination. This new insight, though only tested here using the error-clamp paradigm, can be further utilized in other domains, e.g., conventional visuomotor rotation or force field adaptation. We hope this serves as an initial endeavor in developing some computational models for proprioception studies. Please see the extended discussion on this matter in the revision.

**Recommendations for the authors:**
Revisions:All three reviewers were positive about the work and have provided a set of concrete and well-aligned suggestions, which the authors should address in a revised version of the article. These are listed below.A few points of particular note:(1) There are a lot of discussions about the actual interpretation of behavioral data from this paper or past papers (Tsay et al. 2020 or 2021) and how it matches the different variables of the model.(2) There are some discussions on the results of the first experiment, both in terms of how it is reported (providing degrees of visual angle) and how it is different than previous results (importance of the point of fixation). We suggest also discussing a few papers on eye movements during motor adaptation from the last years (work of Anouk de Brouwer and Opher Donchin). Could the authors also discuss why they found opposite results to that of previous visual uncertainty studies (i.e., visual uncertainty attenuates learning with large, but not small, visual errors); rather than the other way around as in Burge et al and Tsay et al 2021 and Makino Nozaki 2023 (where visual uncertainty attenuates small, but not large, visual errors).(3) It is recommended by several reviewers to discuss the applicability of the model to other areas/perturbations.(4) Several reviewers and I believe that the impact of the paper would be much higher if the code to reproduce all the simulations of the model is made available to the readers. In addition, while I am very positive about the fact that the authors shared the data of their experiments, metadata seems to be missing while they are highly important because these data are otherwise useless.

Thank you for the concise summary of the reviewers’ comments. We have addressed their concerns point by point.

**Reviewer #2 (Recommendations For The Authors):**
L142: The linear increase in visual uncertainty should be substantiated by previous research in vision science. Please cite relevant papers and discuss why the linear model is considered reasonable.

We cited relevant studies in vision science. Their focus is more about eccentricity inflate visual uncertainty, similar to our findings that deviations from the fixation direction inflate visual uncertainty about motion direction.

We also want to add that our model performance does not hinge on a strict linear function of visual uncertainty. Say, if it is a power function with an increasing slope, our model will still predict the major findings presented in the paper. It is the increasing trend of visual uncertainty, which is completely overlooked by previous studies, that lead to various seemingly puzzling findings in implicit adaptation. Furthermore, without assuming a linear function, we fitted the large dataset of motor adaptation from Exp2 to numerically estimate the visual uncertainty. This estimated visual uncertainty has a strong linear relationship with perturbation size (R = 0.991, p<0.001). In fact, the model-fitted visual uncertainty is very close to the values we obtained in Exp1. We now included this new analysis in the revision. See details in Supplementary text 2 and Figure S7.

L300: I found it challenging to understand the basis for this conclusion. Additional explanatory support is required.

We unpacked this concluding sentence as follows:

“The observed proprioceptive bias is formally modeled as a result of the biasing effect of the perceived hand estimate x_hand_hat. In our mini-block of passive localization, the participants neither actively moved nor received any cursor perturbations for three trials in a row. Thus, the fact that the measured proprioceptive bias is reduced to nearly zero at the third trial suggests that the effect of perceived hand estimate x_hand_hat decays rather rapidly.”

L331: For the general reader, a visual representation of what the blurring mask looks like would be beneficial.

Thanks for the nice suggestion. We added pictures of a clear and a blurred cursor in Figure 5D.

L390: This speculation is intriguing. It would be helpful if the authors explained why they consider causal inference to operate at an explicit process level, as the reasoning is not clear here, although the idea seems plausible.

Indeed, our tentative conclusion here is only based on the model comparison results here. It is still possible that causal inference also work for implicit adaptation besides explicit adaptation. We make a more modest conclusion in the revision:

“The casual inference model is also based on Bayesian principle, then why does it fail to account for the implicit adaptation? We postulate that the failure of the causal inference model is due to its neglect of visual uncertainty as a function of perturbation size, as we revealed in Experiment 1. In fact, previous studies that advocating the Bayesian principle in motor adaptation have largely focused on experimentally manipulating sensory cue uncertainty to observe its effects on adaptation (Burge et al., 2008; He et al., 2016; Körding & Wolpert, 2004; Wei & Körding, 2010), similar to our Experiment 4. Our findings suggest that causal inference of perturbation alone, without incorporating visual uncertainty, cannot fully account for the diverse findings in implicit adaptation. The increase in visual uncertainty by perturbation size is substantial: our Experiment 1 yielded an approximate seven-fold increase from a 4° perturbation to a 64° perturbation. We have attributed this to the fact that people fixate in the desired movement direction during movements. Interestingly, even for conventional visuomotor rotation paradigm where people are required to “control” the perturbed cursor, their fixation is also on the desired direction, not on the cursor itself (de Brouwer, Albaghdadi, et al., 2018; de Brouwer, Gallivan, et al., 2018). Thus, we postulate that a similar hike in visual uncertainty in other “free-viewing” perturbation paradigms. Future studies are warranted to extend our PEA model to account for implicit adaptation in other perturbation paradigms.”

L789: The method of estimating Sigma_hand in the brain was unclear. Since Bayesian computation relies on the magnitude of noise, the cognitive system must have estimates of this noise. While vision and proprioception noise might be directly inferred from signals, the noise of the hand could be deduced from the integration of these observations or an internal model estimate. This process of estimating noise magnitude is theorized in recursive Bayesian integration models (or Kalman filtering), where the size estimate of the state noise (sigma_hand) is updated concurrently with the state estimate (x_hand hat). The equation in L789 and the subsequent explanation appear to assume a static model of noise estimation. However, in practice, the noise parameters, including Sigma_hand, are likely dynamic and updated with each new observation. A more detailed explanation of how Sigma_hand is estimated and its role in the cognitive process.

This is a great comment. In fact, if a Kalman filter is used, the learning rate and the state noise all should be dynamically updated on each trial, under the influence of the observed (x_v). In fact, most adaptation models assume a constant learning rate, including our model here. But a dynamic learning rate (B in our model) is something worth trying. However, in our error-clamp setting, x_v is a constant, thus this observation variable cannot dynamically update the Kalman filter; that’s why we opt to use a “static” Bayesian model to explain our datasets. Thus, Sigma_hand can be estimated by using Bayesian principles as a function of three cues available, i.e., the proprioceptive cue, the visual cue, and the motor prediction cue. We added a

detailed derivation of sigma_hand in the revision in Supplementary text 1.

**Reviewer #3 (Recommendations For The Authors):**
We observed values in Fig 2C for the 64-degree perturbation that seem to be outliers, i.e., greater than 50 degrees. It is unclear how a psychometric curve could have a "slope" or JNP of over 60, especially considering that the tested range was only 60. Since the data plotted in panel C is a collapse of the signed data in panel B, it is perplexing how such large data points were derived, particularly when the signed uncertainty values do not appear to exceed 30.Related to the previous point, we would also recommend connecting individual data points: if the uncertainty increases (linearly or otherwise), then people with low uncertainty at the middle distance should also have low uncertainty at the high distance, and people with high uncertainty at one point, should also have that at other distances. Or perhaps the best way to go about this is to use the uncertainty at the two smaller perturbations to predict uncertainty at the largest perturbation for each participant individually?

Thank you for your suggestion to examine the consistency of individual levels of visual uncertainty across perturbation sizes. First, a sigma_v of 60 degrees is well possible, naturally falling out of the experimental data. It shows some individuals indeed have large visual uncertainty. Given these potential outliers (which should not be readily removed as we don’t have any reason to do so), we estimated the linear function of sigma_v with a robust method, i.e., the GLM with a gamma distribution, which favors right-skewed distribution that can well capture positive outliers. Furthermore, we added in our revision a verification test of our estimates of sigma_v: we used Exp2’s adaptation data to estimate sigma_v without assuming its linear dependency. As shown, the model-fitted sigma_v closely matched the estimated ones from Exp1 (see Supplementary text 2 and Figure S7).

We re-plotted the sigma_v with connected data points provided, and the data clearly indicate that individuals exhibit consistent levels of visual uncertainty across different perturbation sizes, i.e. those with relatively lower uncertainty at middle distances (in fact, angles) tend to exhibit relatively lower uncertainty at higher distances too, and similarly, those with higher uncertainty at one distance maintain that level of uncertainty at other distances. This is confirmed by spearman correlation analysis to assess the consistency of uncertainties across different degrees of perturbation among individuals. Again, we observed significant correlations between perturbation angles, indicating good individual consistency (4 and 16 degrees, rho = 0.759, p<0.001; 16 and 64 degrees, rho = 0.527, p = 0.026).

**Author response image 4. sa4fig4:** 

The illustration in Fig 2A does not seem to show a stimulus that is actually used in the experiment (looks like about -30{degree sign} perturbation). It would be good to show all possible endpoints with all other visual elements to scale - including the start-points of the PEST procedure.

Thanks for the suggestion. We updated Fig 2A to show a stimulus of +16 degree, as well as added an additional panel to show all the possible endpoints.

Finally (related to the previous point), in lines 589-591 it says the target is a blue cross. Then in lines 614-616, it says participants are to fixate the blue cross or the start position. The start position was supposed to have disappeared, so perhaps the blue plus moved to the start position (which could be the case, when looking at the bottom panel in Fig 2A, although in the illustration the plus did not move fully to the start position, just toward it to some degree). Perhaps the descriptions need to be clarified, or it should be explained why people had to make an eye movement before giving their judgments. And if people could have made either (1) no eye movement, but stayed at fixation, (2) moved to the blue plus as shown in the last panel in Fig 2A, or (3) fixated on the home position, we'd be curious to know if this affected participants' judgments.

Thanks for pointing that out. The blue cross serves as the target in the movement task, then disappears with the cursor after 800ms of frozen time. The blue cross then appeared in the discrimination task at the center of the screen, i.e. the start location. Subjects were asked to fixate at the blue cross during the visual discrimination task. Note this return the fixation to the home position is exactly what we will see in typical error-clamp adaptation: once the movement is over, people guided their hand back to the home position. We performed a pilot study to record the typical fixation pattern during error-clamp adaptation, and Exp1 was intentionally designed to mimic its fixation sequence. We have now updated the description of Figure 2A, emphasizing the stimulus sequence. .

In Figure 4A, the label "bias" is confusing as that is used for recalibrated proprioceptive sense of hand position as well as other kinds of biases elsewhere in the paper. What seems to be meant is the integrated hand position (x-hat_hand?) where all three signals are apparently combined. The label should be changed and/or it should be clarified in the caption.

Thanks for pointing that out, it should be x_hand_hat, and we have corrected this in the revised version of Figure 4.

In the introduction, it is claimed that larger perturbations have not been tested with "implicit adaptation" paradigms, but in the same sentence, a paper is cited (Moorehead et al., 2017) that tests a rotation on the same order of magnitude as the largest one tested here (95{degree sign}), as well as much larger rotations (135{degree sign} and 175{degree sign}). With error-clamps. Interestingly, there is no adaptation in those conditions, which seems more in line with the sensory cue integration model. Can the PEA model explain these results as well? If so, this should be included in the paper, and if not, it should be discussed as a limitation.

First, we double checked our manuscript and found that we never claimed that larger perturbations had not been tested.

We agree that it is always good to have as many conditions as possible. However, the 135 and 175 degree conditions would lead to minimum adaptation, which would not help much in terms of model testing. We postulated that this lack of adaptation is simply due to the fact that people cannot see the moving cursor, or some other unknown reasons. Our simple model is not designed to cover those kinds of extreme cases.

Specify the size of the arc used for the proprioceptive tests in Exp 3 and describe the starting location of the indicator (controlled by the left hand). Ideally, the starting location should have varied across trials to avoid systematic bias.

Thank you for the comments. The size of the arc used during these tests, as detailed in the methods section of our paper, features a ring with a 10 cm radius centered at the start position. This setup is visually represented as a red arc in Figure 7B.

After completing each proprioceptive test trial, participants were instructed to position the indicator at approximately -180° on the arc and then relax their left arm. Although the starting location for the subsequent trial remained at-180°, it was not identical for every trial, thereby introducing slight variability.

Please confirm that the proprioceptive biases plotted in Fig 4E are relative to the baseline.

Thank you for bringing this to our attention. Yes, the proprioceptive biases illustrated in Figure 4E are indeed calculated relative to the baseline measurements. We have added this in the method part.

Data availability: the data are available online, but there are some ways this can be improved. First, it would be better to use an open data format, instead of the closed, proprietary format currently used. Second, there is no explanation for what's in the data, other than the labels. (What are the units? What preprocessing was done?) Third, no code is made available, which would be useful for a computational model. Although rewriting the analyses in a non-proprietary language (to increase accessibility) is not a reasonable request at this point in the project, I'd encourage it for future projects. But perhaps Python, R, or Julia code that implements the model could be made available as a notebook of sorts so that other labs could look at (build on) the model starting with correct code - increasing the potential impact of this work.

Great suggestions. We are also fully supportive of open data and open science. We now:

(1) Updated our data and code repository to include the experimental data in an open data format (.csv) for broader accessibility.

(2) The data are now accompanied by detailed descriptions to clarify their contents.

(3) We have made the original MATLAB (.m) codes for data analysis, model fitting and simulation available online.

(4) We also provide the codes in Jupyter Notebook (.ipynb) formats.

These updates can be found in the revised “Data Availability” section of our manuscript.

References

Bromberg, Z., Donchin, O., & Haar, S. (2019). Eye Movements during Visuomotor Adaptation Represent Only Part of the Explicit Learning. *eNeuro*, *6*(6). https://doi.org/10.1523/ENEURO.0308-19.2019

Burge, J., Ernst, M. O., & Banks, M. S. (2008). The statistical determinants of adaptation rate in human reaching. *Journal of Vision*, *8*(4), 1–19.

de Brouwer, A. J., Gallivan, J. P., & Flanagan, J. R. (2018). Visuomotor feedback gains are modulated by gaze position. *Journal of Neurophysiology*, *120*(5), 2522–2531.

Egly, R., & Homa, D. (1984). Sensitization of the visual field. *Journal of Experimental Psychology. Human Perception and Performance*, *10*(6), 778–793.

Kim, H. E., Parvin, D. E., & Ivry, R. B. (2019). The influence of task outcome on implicit motor learning. *eLife*, *8*. https://doi.org/10.7554/eLife.39882

Klein, S. A., & Levi, D. M. (1987). Position sense of the peripheral retina. *JOSA A*, *4*(8), 1543–1553.

Levi, D. M., Klein, S. A., & Yap, Y. L. (1987). Positional uncertainty in peripheral and amblyopic vision. *Vision Research*, *27*(4), 581–597.

Makino, Y., Hayashi, T., & Nozaki, D. (2023). Divisively normalized neuronal processing of uncertain visual feedback for visuomotor learning. *Communications Biology*, *6*(1), 1286.

Owsley, C., Ball, K., & Keeton, D. M. (1995). Relationship between visual sensitivity and target localization in older adults. *Vision Research*, *35*(4), 579–587.

Simani, M. C., McGuire, L. M. M., & Sabes, P. N. (2007). Visual-shift adaptation is composed of separable sensory and task-dependent effects. *Journal of Neurophysiology*, *98*(5), 2827–2841.

Tsay, J. S., Avraham, G., Kim, H. E., Parvin, D. E., Wang, Z., & Ivry, R. B. (2021). The effect of visual uncertainty on implicit motor adaptation. *Journal of Neurophysiology*, *125*(1), 12–22.

Tsay, J. S., Chandy, A. M., Chua, R., Miall, R. C., Cole, J., Farnè, A., Ivry, R. B., & Sarlegna, F. R. (2024). Minimal impact of proprioceptive loss on implicit sensorimotor adaptation and perceived movement outcome. *bioRxiv : The Preprint Server for Biology*. https://doi.org/10.1101/2023.01.19.524726

Tsay, J. S., Kim, H., Haith, A. M., & Ivry, R. B. (2022). Understanding implicit sensorimotor adaptation as a process of proprioceptive re-alignment. *eLife*, *11*, e76639.

Wei, K., Stevenson, I. H., & Körding, K. P. (2010). The uncertainty associated with visual flow fields and their influence on postural sway: Weber’s law suffices to explain the nonlinearity of vection. *Journal of Vision*, *10*(14), 4.

White, J. M., Levi, D. M., & Aitsebaomo, A. P. (1992). Spatial localization without visual references. *Vision Research*, *32*(3), 513–526.